



# Driving mechanisms of organic carbon burial in the Early Cretaceous South Atlantic Cape Basin (DSDP Site 361)

Wolf Dummann[1,a], Sebastian Steinig[2,b], Peter Hofmann[1], Matthias Lenz[1], Stephanie Kusch[1], Sascha Flögel[2], Jens Olaf Herrle[3], Christian Hallmann[4,5], Janet Rethemeyer[1], Haino Uwe Kasper[1], Thomas Wagner[6]

[1]Institute of Geology and Mineralogy, University of Cologne, Cologne, D-50674, Germany
[2]GEOMAR Helmholtz Centre for Ocean Research Kiel, Kiel, D-24148 Kiel, Germany
[3]Institute of Geosciences, Goethe-University Frankfurt, Frankfurt am Main, D-60438, Germany
[4]Max Planck Institute for Biogeochemistry, Jena, D-07745, Germany
[5]MARUM, University of Bremen, Bremen, D-28359, Germany
[6]Lyell Centre, School of Energy, Geoscience, Infrastructure and Society, Heriot-Watt University, Edinburgh, EH14 4AS, UK
[a]now at: Institute of Geosciences, Goethe-University Frankfurt, Frankfurt am Main, D-60438, Germany
[b]now at: School of Geographical Sciences, University of Bristol, Bristol, BS8 1SS, UK

*Correspondence to*: Wolf Dummann (wdummann@uni-koeln.de)

**Abstract.** Extensive black shale deposits formed in the Early Cretaceous South Atlantic, supporting the notion that this emerging ocean basin was a globally important site of organic carbon burial. The magnitude of organic carbon burial in marine basins is known to be controlled by various tectonic, oceanographic, hydrological, and climatic processes acting on different temporal and spatial scales, the nature and relative importance of which are poorly understood for the young South Atlantic. Here we present new bulk and molecular geochemical data from an Aptian–Albian sediment record recovered from the deep Cape Basin at Deep Sea Drilling Project (DSDP) Site 361, which we combine with general circulation model results to identify driving mechanisms of organic carbon burial. A multi-million year decrease (i.e. Early Aptian–Albian) in organic carbon burial, reflected in a lithological succession of black shale, gray shale, and red beds, was caused by increasing bottom water oxygenation due to abating tectonic restriction via South Atlantic-Southern Ocean gateways. These results emphasize basin evolution and ocean gateway development as a decisive primary control on enhanced organic carbon preservation in the Cape Basin at geological time scales (>1 Myr). The Early Aptian black shale sequence comprises alternations of shales with high (>5%) and relatively low (~3%) organic carbon content of marine sources, the former being deposited during the global Oceanic Anoxic Event (OAE) 1a, as well as during repetitive events before and after OAE 1a. In all cases, these short-term events of enhanced organic carbon burial coincided with strong influxes of sediments derived from the proximal African continent, indicating closely coupled climate–land–ocean interactions. Supported by our model results, we propose that fluctuations in weathering-derived nutrient input from the southern African continent, linked to fluctuations in $p$CO2 and/or orbitally driven humidity/aridity, were the underlying drivers of short-term organic carbon burial in the deep Cape Basin. These results suggest that deep marine environments of emerging ocean basins responded sensitively and directly to short-term fluctuations in riverine nutrient fluxes. We explain this relationship by the lack of wide and mature continental shelf seas that could have acted as a barrier or filter for nutrient transfer from the continent into the deep ocean.

## 1   Introduction

The Early Cretaceous epoch (~145–100.5 Ma) has long been a focus of interest for geologists and the petroleum industry due to the widespread occurrence of marine black shales (i.e. organic carbon rich, fine-grained sediments), which were preferentially deposited in emerging ocean basins that developed in the wake of the breakup of Pangaea (Stein et al., 1986; Zimmerman et al., 1987). Within these basins, organic carbon (OC) burial was controlled by a complex interplay of productivity, preservation, and dilution (Pedersen and Calvert, 1990; Tyson, 2005; Arthur and Sageman, 1994; Bralower and Thierstein, 1984), which was affected by multiple processes acting on different time scales. On a multi-million year





(geological) time scale, OC burial was controlled by tectonic processes modifying basin geometry, marine gateway evolution, and ocean circulation (Donnadieu et al., 2016; Dummann et al., 2020; Arthur and Natland, 1979; Wagner and Pletsch, 1999), while shorter term (e.g. orbitally driven) changes in oceanic and atmospheric circulation triggered pronounced fluctuations in

the magnitude of OC burial on time scales of ten to hundred thousand years (Beckmann et al., 2005; Behrooz et al., 2018; Hofmann and Wagner, 2011; Wagner et al., 2013; Kolonic et al., 2005; Herrle et al., 2003). During brief episodes ≤~1 Myr, marine black shales were deposited regionally to globally and accompanied by intense perturbations in ocean chemistry and ecology (Jenkyns, 2010; Weissert et al., 1998; Erba, 1994). These carbon–climate perturbation events are described as Oceanic Anoxic Events (OAEs; Schlanger and Jenkyns (1976)). OAE 1a, one of the most severe OAEs (Jenkyns, 2010 and references

therein), occurred during the Early Aptian (Coccioni et al., 1992) and was marked by a distinct negative stable carbon isotope excursion at its onset (Menegatti et al., 1998), evidence for rapid warming (O'Brien et al., 2017 and references therein), and elevated atmospheric $CO_2$ concentrations (Naafs et al., 2016).

During the Early Aptian, at the time of OAE 1a, the emerging South Atlantic Ocean consisted of two main rift basins that progressively opened from South to North (Pérez-Díaz and Eagles, 2017) and which were separated from each other by the

volcanic Rio Grande Rise/Walvis Ridge (Figure 1). Each of the two sub-basins acted as a major depocenter for OC-rich sediments (Zimmerman et al., 1987; Arthur and Natland, 1979; Macdonald et al., 2003) with exceptionally high OC burial rates in the Early Cretaceous, accounting for around 16% of global excess OC burial in an area that covered only around 1% of the total global ocean (McAnena et al., 2013). Data from Deep Sea Drilling Project (DSDP) Site 364, located at a paleo-latitude of 25° S and an estimated paleo-water depth of ~300–400 m (Zimmerman et al., 1987) in the Angola Basin (Figure

1), suggest that enhanced OC deposition occurred north of the Rio Grande Rise/Walvis Ridge during the Early to early Late Aptian, sustained by the restricted basin geometry and the continuous supply of nutrients from surrounding continents (Arthur and Natland, 1979; Behrooz et al., 2018; Naafs and Pancost, 2014). In line with previous studies (e.g. Beckmann et al., 2005), these data indicate that marine OC burial dynamics at tropical latitudes were tightly coupled to continental hydrology and run-off variability beneath the ascending limb of the atmospheric Hadley Cell (the Intertropical Convergence Zone; ITCZ). Near-

shore sediment records located along continental margins, such as DSDP Site 364, often show particularly pronounced cycles of variable OC quantity and quality, as this is where hydrological changes on the continent are most directly translated into marine sediments, while less pronounced cycles at sites further offshore reflect an attenuation of the signal with increasing water depth and distance from the continent (Wagner et al., 2013).

Our study investigates the driving mechanisms of marine OC burial in the deep Cape Basin, south of the Rio Grande

Rise/Walvis Ridge, and their connections to mid-latitude atmospheric circulation and continental hydrology, which have been studied less extensively. We provide new inorganic and organic (bulk and molecular) geochemical data from DSDP Site 361, located at a paleo-latitude of ~45° S and a paleo-water depth of 2–2.5 km (Figure 1; Melguen (1978); Thiede and Van Andel (1977); Van Andel et al. (1977)), where an extensive Aptian black shale sequence was recovered (The Shipboard Scientific Party, 1978), including OAE 1a (Dummann et al., 2020). These black shales show repetitive alternations of OC-rich and

(relatively) OC-poor sediments (Dummann et al., 2020), indicating highly dynamic depositional conditions on a time scale of <1 Myr. We combine our new proxy data with general circulation model data to provide insights into the Early Cretaceous variability of atmospheric circulation and continental hydrology at mid-latitudes and to evaluate its importance for marine OC burial in the deep Cape Basin.

## 2    Materials and methods

### 2.1    DSDP Site 361

Site 361 was drilled on the lower continental rise of South Africa (35°03.97' S, 15°26.91' E) at a modern water depth of 4549 m (The Shipboard Scientific Party, 1978). The recovered 390 m long Early Cretaceous sediment sequence mainly consists of



Early Aptian black shales intercalated with turbiditic sand- and siltstones (Figure 2a), which were deposited as part of a deep proximal fan to fan-valley environment (Natland, 1978). Early Aptian sediments are overlain by Late Aptian to Albian gray
shales and reddish marlstones (Figure 2a).

Total organic carbon (TOC) and bulk $\delta^{13}C_{org}$ data at Site 361 (Figure 2b,c) were previously reported by Dummann et al. (2020). These data show that the sediment sequence comprises a stratigraphic succession of three lithological units: Early Aptian black shales with TOC contents generally exceeding 3%, Late Aptian gray shales with TOC contents between 0.5 and 3%, and Albian red beds largely devoid of OC (Figure 2b). Early Aptian black shales at Site 361 predominantly show background TOC
values of ~3%, which episodically increase to >5% TOC and up to ~20%. One of these high-TOC black shale intervals (~13% TOC) corresponds to the global OAE 1a, which has been identified based on calcareous nannofossil biostratigraphy and $\delta^{13}C_{org}$ stratigraphy (Figure 2a,b; Dummann et al. (2020)).

## 2.2  Analytical methods

### 2.2.1  Major and trace element analysis by XRF

Major (Al, Si, K, Ti) and trace element (Zr, Ni, Cu, V, Zn) concentrations were determined on the complete sample set (*n*=131) using an Itrax x-ray fluorescence (XRF) Core Scanner (Cox Analytical Systems, Sweden) equipped with a Cr-He tube. Dried and ground samples were pressed into plastic cubes, which were aligned under the core scanner. Analyses were performed at a resolution of 1 mm and an integration time of 60 s per measurement, yielding nine to twelve individual measurements per sample, which were averaged over each sample. Absolute element concentrations were quantified by calibration against in-
house standards of known elemental composition, which were measured alongside the samples (for details see Supplement S1).

### 2.2.2  Trace element analysis by ICP-MS

In addition to XRF analysis, 15 Early Aptian black shale samples were investigated for their immobile trace element (Th, Sc, Zr) and rare earth element (REE) composition, which was determined using a SCIEX ELAN 6100 inductively coupled plasma
mass spectrometer (ICP-MS; Perkin Elmar, USA) following total acid digestion. Aliquots of dried and ground samples were combusted at 550 °C for 4 h to remove OC and subsequently decomposed in a 6 AM pressure digestion system (PDS6, Loftfields Analytical Solutions, Germany), following published protocols (Dulski, 2001). Acids used for digestion were HF (40%, suprapure, Merck, Germany), $HNO_3$ (65%, suprapure, Merck, Germany), $HClO_4$ (70%, ultrapure, VWR, USA), and HCl (30%, suprapure, Merck, Germany). Quantification was conducted using two external multi-element standard solutions
(1 ng$^{-1}$, 10 ng$^{-1}$) and a matrix solution. An internal Ru-Re standard was analyzed alongside the samples to monitor instrumental drift. Two certified reference materials (CRM) JA-2 (Dulski, 2001) and GA (Govindaraju, 1994) were used to monitor precision and accuracy. All standards reproduced within ±3%.

### 2.2.3  Lipid biomarker analysis

Aliquots from a total of 72 samples (1–8 g) were extracted with methanol (Merck SupraSolv®, Germany; 30 mL), methanol–
dichloromethane (1:1, v:v; 30 mL), and dichloromethane (Merck SupraSolv®, Germany; 30 mL) for 10 minutes each in an ultrasonic bath. The extracts were combined in a separatory funnel and mixed with ultrapure water to remove methanol. The dichloromethane fraction was decanted and dried under mild vacuum (40 °C, 800 mbar) using a rotary evaporator. Acid-activated copper turnings were added to the extract to remove elemental sulfur. The samples were saponified using a 0.5 molar potassium hydroxide solution (methanol–water 9:1, v:v). Neutral lipids were extracted from the saponification solution with
hexane (Merck SupraSolv®, Germany) and partitioned into three polarity fractions (aliphatic hydrocarbons, aromatic hydrocarbons, and heteroatomic (NSO) compounds) over a self-packed silica column (500 mg, 0.063–0.200 mm, 70–230 mesh, Merck, Germany; deactivated with 1% ultrapure water) using hexane, hexane–dichloromethane (2:1, v:v), and



dichloromethane–methanol (1:1, v:v), respectively. To reduce interferences during mass spectrometric analysis, the aliphatic fraction was further separated into saturated and unsaturated compounds using silver-ion chromatography (10% AgNO$_3$-coated

silica gel, Sigma-Aldrich, USA), which were eluted with hexane and ethyl acetate (Merck SupraSolv®, Germany), respectively. $d_{50}$-Tetracosane and $d_4$-cholestane were used as internal standards and were added to the saturated aliphatic fraction prior to analysis.

The saturated aliphatic fractions were analyzed using coupled gas chromatography-mass spectrometry (GC-MS) and GC-MS/MS. The distribution of acyclic hydrocarbons (i.e. $n$-alkanes, acyclic isoprenoids) were analyzed using an Agilent 6890N

GC coupled to an Agilent 5975 MSD operated in selected ion monitoring (SIM) mode scanning for $m/z$ 85 ([C$_6$H$_{13}$]$^+$) and $m/z$ 98 ([C$_6$D$_{13}$]$^+$) fragments. Samples were injected onto a DB5-MS column (50 m, 0.2 mm, 0.25 µm) using a split/splitless injector operated in splitless mode and heated from 70 °C to 250 °C. Helium was used as carrier gas at a constant flow rate of 1.3 mL min$^{-1}$. The GC temperature was held at 40 °C for 2 min, increased to 140 °C at 10 °C min$^{-1}$ and to 320 °C at 5 °C min$^{-1}$. The final temperature was held for 22 min. Ionization was achieved by electron impact (EI) at 70 eV and 230ºC. Reproducibility

was monitored using an authentic $n$-alkane standard and was better than 1%.

Sterane and hopane distributions were analyzed using an Agilent 7890B GC coupled to an Agilent 7000C EI-TQ-MS operated in multiple reaction monitoring (MRM) mode. Samples were injected onto a HP5-MS column (50 m, 0.25 mm, 0.25 µm) using a split/splitless injector operated in splitless mode at 290 °C. Helium was used as carrier gas at a constant flow rate of 1.2 mL min$^{-1}$. The initial GC temperature was held at 60 °C for 2 min, then increased to 140 °C at 10 °C min$^{-1}$ and to 320 °C at 4 °C

min$^{-1}$, which was held for 15 min. Ionization was achieved by EI at 70 eV and 300ºC. Argon was used as a collision gas at a collision energy of 10 V. Reproducibility was monitored using an in-house standard containing all targeted compounds at similar concentration levels. The reproducibility of measurements varied between different compounds and was mostly better than 5%, but always better than 10% for all reported biomarker ratios.

### 2.3   Sediment provenance assessment

Immobile trace element and REE geochemistry were used to constrain the provenance of Early Aptian black shales. To define provenance types and tectonic settings, we used discrimination plots following the approaches of Bhatia and Crook (1986) and McLennan et al. (1993). Potential sediment source regions were identified based on comparison with geochemical data compiled from the literature. Given that our study focusses on mudstone intervals at Site 361, we only report data for shale and mudstone samples from sediment source regions, while data for sandstones were omitted. This data compilation including

all references is provided in Supplement S2 and S3.

### 2.4   General circulation modeling

Model simulations were carried out with the Kiel Climate Model (KCM; Park et al. (2009)). The KCM uses the ECHAM5 spectral atmospheric model with a horizontal resolution of ~2.8° (T42) on 19 vertical levels (Roeckner et al., 2003) coupled to the ocean–sea ice component NEMO (Madec, 2008) on a tripolar grid with a horizontal resolution of 2° (ORCA2) and 31

vertical levels. The meridional ocean resolution gradually increases to 0.5° towards the equator. Modifications to the model for the Early Cretaceous configuration are described in Dummann et al. (2020). The only difference to this study is that the previously used Early Albian paleobathymetry of the study area was replaced by the respective Early Aptian reconstruction (Sewall et al., 2007) north of the Walvis Ridge/Rio Grande Rise (~30° S) to account for the reduced northward extent of the Angola Basin during the Aptian.

Two 3000 year-long simulations with atmospheric $p$CO$_2$ levels of 600 ppm and 1200 ppm were performed to match the published range of stomata-based $p$CO$_2$ reconstructions for the Aptian–Albian (Jing and Bainian, 2018). Even though peak $p$CO$_2$ concentrations during some parts of OAE 1a may have been even higher (Naafs et al., 2016), we interpret a $p$CO$_2$ of



1200 ppm to more closely represent average $p$CO$_2$ levels during OAE 1a. Results were averaged over the last 100 model years, while the globally depth-integrated temperature drift for both simulations over the last 1000 model years was below 0.2 °C.

## 3   Results

### 3.1   Total organic carbon content

Aptian–Albian sediments at Site 361 range from 0 to 19.7% TOC (Figure 2c; Dummann et al. (2020)). Four different lithology types can be discriminated based on sediment color and TOC content: Red beds with less than 0.1% TOC present in core 26, gray shales with 1–3% TOC in core 27, and black shales with more than 3% TOC in core 28 and below. Black shales can be further subdivided into "low-TOC black shales" (i.e. 3–5% TOC), which constitute the background sedimentation in cores 28–48 (Figure 2a) and "high-TOC black shales" (i.e. >5% TOC), which appear to be restricted to narrow stratigraphic intervals. High-TOC black shale intervals occur during OAE 1a, but also at other stratigraphic horizons below and above OAE 1a.

### 3.2   Sediment composition and provenance

#### 3.2.1   Distribution of K, Si, and Zr

To track changes in geochemical composition related to grain size and weathering state of the sediments, we report K/Al, Si/Al and Zr/Al ratios obtained by XRF analysis (Figure 2d–f). All black shales at Site 361 show Si/Al and Zr/Al ratios typical for fine-grained sediments with ratios close to average shale (AS) values of 3.11 and 18, respectively (Wedepohl, 2004, 1971). Neither Si/Al nor Zr/Al ratios differ substantially between high-TOC and low-TOC black shale intervals (Figure 2e). In contrast, K/Al ratios vary in concert with TOC (Figure 2d) with high-TOC black shales showing a higher mean K/Al ratio of 0.38 ± 0.04 ($\bar{x} \pm 1\sigma$) compared to low-TOC black shales with a mean ratio of 0.32 ± 0.04.

#### 3.2.2   Sediment provenance

Black shale samples show distinct trace element and REE patterns typical for continental crust with clear differences between low-TOC and high-TOC black shales (Figure 3c–e). Low-TOC black shales show typical characteristics for a recycled sedimentary rock provenance (McLennan et al., 1993), including Th/Sc >1 and enrichment of Zr/Sc relative to Th/Sc (Figure 3c). In addition, all low-TOC samples show Th/Zr and La/Sc ratios indicative for an active continental margin signature (Figure 3d), implying a felsic to intermediate sediment source composition, which is also supported by a strong enrichment of the light REE La, Ce, Pr, Nd, and Sm and a pronounced negative Eu anomaly (Figure 3e).

The trace element and REE signature of high-TOC black shales suggests a higher contribution from mafic sediment sources, as indicated by slightly lower Th/Sc ratios (Figure 3c), lower La/Sc ratios (Figure 3d) and depletion of light REE compared to low-TOC black shales (Figure 3e). High-TOC samples plot close to the continental island arc endmember (Figure 3d), indicating a bulk sediment source composition close to granodiorite.

### 3.3   Inorganic paleo-redox parameters

#### 3.3.1   Distribution of sulfur and iron

The stratigraphic variation of sulfur content closely parallels that of the TOC content (Figure 4b). Sulfur contents of Early Aptian black shales range from 1.3% to 9.5% and decrease to mean values of 0.4% ± 0.5% in Late Aptian gray shales. Albian red beds are essentially sulfur free. Low-TOC black shales are characterized by a mean sulfur content of 2.2 ± 0.5%, while high-TOC black shales contain an almost twice as high mean sulfur content (4.2 ± 1.7%).

Fe-S-TOC relationships reflect changes in paleo-redox conditions (Figure 5; Dean and Arthur (1989) and Arthur and Sageman (1994)). Red bed and gray shale samples scatter around a S/C mixing line of ~0.4, typical for sediments deposited in an oxic





environment (Berner and Raiswell, 1983; Leventhal, 1983) and show S/Fe well below a value of 0.45, which is generally used to discriminate oxic from "restricted" (i.e. suboxic to anoxic) conditions (Raiswell et al., 1988). Black shales show overall higher S/Fe ratios, covering a broader range between the 0.25 and the pyrite line. Low-TOC black shales and high-TOC black shales cluster into two populations with minor overlap. S/Fe ratios of low-TOC black shales fall close to 0.45, with a relatively large scatter between 0.25 and 0.75. High-TOC black shales instead tend to have S/Fe ratios greater than 0.45, often exceeding

0.75 (mean of $0.79 \pm 0.27$), indicating "inhospitable" (i.e. strictly anoxic to euxinic) conditions (Raiswell et al., 1988).

### 3.3.2   Distribution of redox-sensitive trace metals

Distribution of the redox-sensitive trace metals (TMs) Ni, Cu, V, and Zn in black shales are presented as TM/Al ratios in Figure 4c–f and enrichment factors (EFs) relative to AS values in Figure 6. All TM/Al ratios co-vary closely with each other and with TOC (Figure 4c–f). Albian red beds all show TM/Al close to or below AS values with mean EFs of $0.7 \pm 0.2$, $0.9 \pm$

0.7, $0.8 \pm 0.5$, and $0.1 \pm 0.2$ for Ni, Cu, V, and Zn, respectively. Similarly, low EFs for V ($1.3 \pm 0.4$) and Zn ($1.8 \pm 1.4$) are present in Late Aptian gray shales, which, however, show substantially higher EFs for Ni ($3.4 \pm 1.4$) and Cu ($3.2 \pm 1.0$). Early Aptian low-TOC black shales show moderate, if any, TM enrichment beyond AS values with mean EFs of $1.2 \pm 1.4$, $1.3 \pm 0.8$, $2.1 \pm 2.0$, and $3.3 \pm 5.5$ for Ni, Cu, V, and Zn, respectively. In contrast, high-TOC black shales are characterized by consistently higher TM EFs, with V and Zn being particularly enriched, as indicated by mean EFs of $9.3 \pm 6.2$ and $27.5 \pm 31.7$ as well as

maximum EFs of 23.2 and 107.1, respectively. Enrichment of Ni and Cu in high-TOC black shales is more modest with mean values of $4.9 \pm 2.6$ and $4.0 \pm 2.2$, respectively.

Cross-plots of TM EFs against TOC provide additional information on the mechanisms of sedimentary TM enrichment and thus paleo-redox conditions (Tribovillard et al., 2006; Algeo and Maynard, 2004). High-TOC black shales show pronounced differences in TM-TOC relationships for Ni, Cu, V, and Zn (Figure 6). Ni and Cu EFs tend to increase with TOC content,

although only Cu exhibits a clear linear relationship ($R^2$=0.38) to TOC content, while V and Zn EFs show a substantial scatter and no relationship to TOC content.

### 3.4   Biomarker distribution

Our biomarker assessment focusses on thermal maturity, OC sources, and paleo-redox conditions, for which we report selected biomarker parameters, while a detailed discussion of whole suite of biomarkers is beyond the scope of this study.

### 3.4.1   Thermal maturity

Sediments at Site 361 are thermally immature at an estimated $R_O$ of <0.4% (Peters et al., 2007). Steranes mainly possess the thermally least stable $5\alpha,14\alpha,17\alpha(H)$-20R or $5\beta,14\alpha,17\alpha(H)$-20R stereoisomeric configurations. Thermally unstable $17\beta,21\beta(H)$-hopanes dominate over more stable $17\alpha,21\beta(H)$ and $17\beta,21\alpha(H)$ hopanes and the most stable configurations such as $17\alpha,21\beta(H)$-30S-homohopanes and $18\alpha(H)$-22,29,30-trisnorneohopane (Ts) are either absent or occur in trace amounts only.

$\beta\beta$-Hopane-ratios ($17\beta,21\beta(H)/(17\alpha,21\beta(H)+17\beta,21\alpha(H)+17\beta,21\beta(H))$-$C_{30}$-hopane) vary between 0.08 and 0.36 (mean of $0.18 \pm 0.06$; Figure 7b), do not increase downcore, and show no systematic trend when compared to TOC, indicating that maturation effects had a negligible impact on down-core variations in the biomarker distribution.

### 3.4.2   Organic carbon sources

Variations in OC sources are assessed based on the relative abundances of $n$-alkanes, desmethylsteranes, and regular hopanes

(i.e. $C_{27}$ and $C_{29}$–$C_{33}$ hopanes). $n$-Alkanes range in chain length from $C_{15}$ to $C_{35}$. High molecular weight (HMW) $C_{25}$ to $C_{35}$ $n$-alkanes show an odd-over-even predominance (with a mean carbon preference index ($CPI_{25-33}$) of 1.9, calculated according to Bray and Evans (1961)), indicating a higher land plant origin (Bray and Evans, 1961; Eglinton and Hamilton, 1967). The ratio





of low molecular weight (LMW) $C_{15}$, $C_{17}$, and $C_{19}$ $n$-alkanes, indicative of algal and/or bacterial OC input (Han and Calvin, 1969), to HMW odd carbon-numbered $C_{25}$ to $C_{35}$ $n$-alkanes varies from 0.2 to 6.7 (Figure 7c). Low-TOC black shales show a

lower mean LMW/HMW ratio of $1.3 \pm 1.3$ compared to high-TOC black shales, which have a mean LMW/HMW ratios of $2.0 \pm 1.3$.

A suite of $C_{27}$ to $C_{35}$ hopanes, reflecting OC inputs from prokaryotic/bacterial sources, and regular $C_{27}$ to $C_{30}$ desmethylsteranes, indicative of eukaryotic OC sources, are present in varying concentrations. Eukaryotic OC generally dominates over bacterial OC, as indicated by sterane/hopane ratios (ratio of $C_{27}$ to $C_{29}$ desmethylsteranes to regular $C_{27}$ and

$C_{29}$–$C_{33}$ hopanes) ranging from 2.3 to 62.5 (Figure 7d). Low-TOC black shales are characterized by mean sterane/hopane ratios of $8.5 \pm 4.7$, which increase in high-TOC black shale interval to a mean value of $28.3 \pm 10.8$.

24-$n$-Propylcholestane (24-npc), a proxy that can indicate contributions of marine algae to OC (Moldowan et al., 1990), is present throughout the record. We normalize the abundance of 24-npc to $C_{29}$-desmethylsterane, which is generally linked to higher land plant inputs (Huang and Meinschein, 1979), to approximate the relative contribution of marine and terrigenous

OC. 24-npc/$C_{29}$-sterane ratios co-vary with TOC (Figure 7e) and range between 0.06 and 0.35. Low-TOC black shales show a 24-npc/$C_{29}$-sterane mean ratio of $0.13 \pm 0.04$, which increases to $0.21 \pm 0.05$ in high-TOC black shales.

Fractional abundances of regular $C_{27}$, $C_{28}$, and $C_{29}$ steranes, depicted as a ternary diagram (Figure 8), are used to differentiate different eukaryotic sources (Huang and Meinschein, 1979). Although low- and high-TOC samples show a great overlap, $C_{29}$-steranes tend to dominate in low-TOC black shales, indicating a relatively high input from terrestrial sources, while $C_{27}$ and

$C_{28}$ dominate over $C_{29}$ in high-TOC black shales, suggesting a greater contribution of marine OC.

### 3.4.3 Paleo-redox conditions

Our paleo-redox assessment is based on the distribution of lycopane, a tail-to-tail linked acyclic isoprenoid, the occurrence of which is limited to anoxic depositional settings (Sinninghe Damsté et al., 2003). Lycopane abundances are normalized to $C_{31}$ $n$-alkane, following Sinninghe Damsté et al. (2003). Lycopane/$n$-$C_{31}$ ratios range from 0.04 to 4.8 and co-vary closely with

TOC (Figure 7f), with low-TOC black shales and high-TOC black shales showing mean ratios of $0.5 \pm 0.4$ and $1.7 \pm 0.8$, respectively. Furthermore, we screened the aromatic hydrocarbon fraction of all samples for isorenieratene derivatives and related compounds, whose presence can be used as a proxy for photic zone euxinia (Koopmans et al., 1996). However, no isorenieratene derivatives were detected in any of the analyzed samples.

## 4 Discussion

The pace of OC burial in the Aptian–Albian Cape Basin was variable. On a multi-million year time scale, OC burial decreased in two steps, as reflected in the lithological succession of black shales in the Aptian (TOC >3%), gray shales in the latest Aptian (0.5–3% TOC), and red beds in the Albian (<0.5% TOC). Superimposed on this long-term trend, we identify several distinct OC burial events, punctuating the Early Aptian interval (Figure 2c). Most of these high-TOC events reach peak TOC levels well above 10%, clearly separating them from background sedimentation with much lower TOC content (~3% TOC).

These high-TOC events were deposited during different time intervals across the Aptian, including, but not limited to, OAE 1a (Figure 2b,c; Dummann et al. (2020)). Geochemical characteristics of all high-TOC events imply comparable depositional conditions with regard to paleo-redox conditions, OC source, and sediment provenance. The limited core recovery, however, hinders the assessment of the duration and the frequency of these short-term events, i.e. they may have lasted longer, and there may have been other high-TOC events, which went unnoticed due to the incomplete stratigraphic coverage. The geochemical

similarities of the Early Aptian high-TOC events including OAE 1a probably suggest comparable forcing-response mechanisms determining the OC sedimentation patterns up to the Late Aptian termination of black shale sedimentation. In order to elucidate potential forcing mechanisms for the short-term OC burial events in the Early Aptian black shale interval,





we develop a depositional concept which can be evaluated with the help of a general circulation model (GCM). The $pCO_2$ set-up of our GCM modeling experiments is geared to represent OAE 1a, as relatively robust information on boundary conditions
are available for this time interval.

### 4.1    Paleo-redox variations

#### 4.1.1    Early Aptian–Albian long-term increase in seawater oxygenation

The Early Aptian–Albian stratigraphic succession consisting of black shales, gray shales, and red beds (Figure 2c) reflects a decrease in OC burial over at least 13 Ma (Gradstein et al., 2012), implying a long-term tectonic mechanism. Previous studies
linked this long-term decrease in OC burial in the Cape Basin to the tectonic opening of the nearby South Atlantic-Southern Ocean gateway (Arthur and Natland, 1979) just south of the study site. New Nd-isotope evidence from various sites from the South Atlantic and the Southern Ocean, including Site 361, combined with ocean current simulations, refined the notion that large-scale deep water circulation changes in response to the progressive opening of two gateways located on the Falkland Plateau and in the Georgia Basin caused the two-step reduction of OC burial at Site 361 via increasing deep water ventilation
(Dummann et al., 2020).

This earlier interpretation by Dummann et al. (2020) is consistent with the new geochemical data presented here, based on (1) sulfur concentrations (Figure 4b), (2) redox-sensitive TM distribution (Figure 4c–f), (3) Fe-S-TOC relationships (Figure 5), and (4) distribution of lycopane (Figure 7f). High abundances of sulfur of >1.4% in both low TOC and high TOC intervals indicate that pyrite formation and/or sulfurization of OC linked to microbial sulfate reduction was a common process
throughout the Aptian, implying anoxic conditions in pore and potentially bottom waters during early diagenesis (Berner, 1970). The degree of pyritization (DOP: ratio of sulfide-bound iron to total reactive iron) depends on the level of oxygen depletion, with DOP ranges of <0.45, 0.45–0.75, and >0.75 reflecting oxic, "restricted" (i.e. suboxic–anoxic), and "inhospitable" (i.e. strictly anoxic–euxinic) conditions, respectively (Raiswell et al., 1988). S/Fe ratios at Site 361 provide a conservative estimate for DOP given that they reflect changes in both reactive and unreactive iron (Raiswell and Canfield,
1998). Accordingly, S/Fe ratios close to or greater than 0.45 in black shales at Site 361 are consistent with suboxic–anoxic bottom water conditions. Occurrences of the acyclic isoprenoid biomarker lycopane have been reported from modern anoxic settings (Sinninghe Damsté et al., 2003; Wakeham et al., 1993) and anoxic depositional settings throughout the geological past (Beckmann et al., 2008; Wagner et al., 2004; Forster et al., 2004; Schouten et al., 2001). Hence, oxygen-deficient bottom water conditions and short post-depositional exposure times to oxygen have been proposed to be a prerequisite for the preservation
of substantial amounts of lycopane (Sinninghe Damsté et al., 2003). Based on these findings, Sinninghe Damsté et al. (2003) introduced the use of lycopane/$n$-$C_{31}$ ratios as a proxy for paleo-oxicity. At Site 361, mean lycopane/$n$-$C_{31}$ ratios of around 0.5 in Aptian low-TOC shales are similar to modern anoxic settings, e.g. oxygen minimum zones (OMZ) along the Peru Margin and the northern Arabian Sea (Sinninghe Damsté et al., 2003). All low-TOC black shales exhibit TM EFs falling close to AS values, indicating a very limited degree of authigenic enrichment. This argues for dysoxic–suboxic (>~0.2 mL $O_2$ L$^{-1}$ $H_2O$)
rather than anoxic conditions (0–0.2 mL $O_2$ L$^{-1}$ $H_2O$) during periods of low-TOC background sedimentation (Algeo and Maynard, 2004).

In contrast to the Early Aptian black shales, geochemical characteristics of Late Aptian gray shales and Early Albian red beds show clear indication of abating anoxia. Evidence includes (1) substantially lower sulfur contents (Figure 4b), (2) S/Fe and S/TOC ratios of <0.45 and ~0.4 (Figure 5), respectively, indicating oxic water column conditions (Raiswell et al., 1988;
Leventhal, 1983), (3) low lycopane/$n$-$C_{31}$ ratios of <0.1 (Figure 7f), and (4) reduction of V and Zn enrichment (Figure 4e–f). These results are consistent and suggest that gray shales and red beds were deposited under permanently oxic water column and near subsurface conditions. Considering above lines of evidence, it seems unlikely that high EFs of Ni and Cu in gray shales (Figure 4c–d) stemmed from redox-related authigenic enrichment processes, but rather suggests additional processes



supplying Ni and Cu to Late Aptian sediments in the Cape Basin, e.g. detrital input (Garver et al., 1996) or scavenging by
biogenic silica (Böning et al., 2015; Twining et al., 2012). The exact mechanism, however, remains to be identified.

### 4.1.2     Short-term variations in paleo-redox conditions during black shale deposition in the Early Aptian

Low-TOC black shales were deposited under dysoxic–suboxic conditions, while anoxia intensified during high TOC intervals
as indicated by (1) higher sulfur contents (Figure 4b), (2) sharp increases in redox sensitive TM/Al ratios (Figure 4c–f), and
(3) elevated lycopane/$n$-$C_{31}$ ratios (Figure 7f). Furthermore, Fe-S-TOC relationships show that ~60% of high-TOC black shales
have S/Fe ratios greater than 0.75 (Figure 5), arguing for strictly anoxic–euxinic conditions (Raiswell et al., 1988).
Distinguishing euxinic conditions, characterized by accumulation of $H_2S$ in the water column, from non-sulfidic but anoxic
('ferruginous') conditions is difficult based on Fe-S-TOC systematics alone. The distribution of redox-sensitive TMs, however,
may help to distinguish these subtle fluctuations at the extreme end of the redox scale (Algeo and Maynard, 2004; Tribovillard
et al., 2006; Meyer and Kump, 2008).

At Site 361, all analyzed redox-sensitive TMs show close parallel trends of enrichment and depletion concurrent with TOC
variations (Figure 4). This strong similarity of trends between different redox-sensitive TMs supports that changes in the
dissolved TM inventory are negligible and that paleo-redox conditions exerted the dominant control over TM enrichment
(Algeo and Maynard, 2008). Cross-plots of TM EFs and TOC reveal two distinct patterns of TM enrichment for Ni and Cu
and V and Zn in high-TOC black shales (Figure 6). Ni and Cu EFs increase in parallel with TOC content, suggesting that
excess TM enrichment was controlled by OC supply to the sediment. Such a coupling of Ni and Cu enrichment with OC flux
has been observed in sediments deposited under both anoxic ferruginous conditions as well as euxinic conditions (Algeo and
Maynard, 2004; Tribovillard et al., 2006) and mainly stems from their behavior as micronutrients (Tribovillard et al., 2006;
Little et al., 2015).

In contrast, high-TOC black shales are characterized by substantial enrichment of V, which is decoupled from OC input, as
indicated by the lack of correlation between V EFs and TOC (Figure 6c). This mode of V enrichment supports euxinic
conditions at the sediment/water interface and possibly the (lower) water column (Algeo and Maynard, 2004; Tribovillard et
al., 2006). The presence of free $H_2S$ favors reduction to V(III) (Wanty and Goldhaber, 1992), which forms insoluble hydroxide-
phases that can precipitate in quantity from the water column and/or the sediment water interface, leading to a decoupling of
V supply from OC flux. Zn EFs exhibit a distribution similar to V EFs (Figure 6e), which is typical for euxinic settings and
potentially results from the formation of independent ZnS phases at the sediment/water interface (Algeo and Maynard, 2004).
Further evidence for euxinic conditions during high-TOC intervals comes from the occurrence of type IIS kerogens (Hartwig
et al., 2012), indicating excess availability of sulfur.

In contrast to many Mesozoic black shales deposited under euxinic conditions (Meyer and Kump, 2008 and references therein),
high-TOC black shales at Site 361, however, do not contain biomarkers derived from phototrophic $H_2S$ oxidizing bacteria (i.e.
isorenieratene derivatives and other aryl isoprenoids, Koopmans et al. (1996)). This general lack of photic zone euxinia-
indicating compounds at Site 361 puts constraints on the extent of water column euxinia and suggests that $H_2S$ did not pervade
into the upper water column. The oceanographic conditions that favored this limited extent of euxinia are difficult to assess
based on the data presented here. However, previous studies indicate that saline, relatively young, and thus probably oxygen-
rich, intermediate water masses entered the Cape Basin from the North (Dummann et al., 2020). These water masses may have
caused an efficient ventilation at intermediate water depths, thereby providing a barrier for upward migration of $H_2S$.

### 4.2     Composition of organic carbon in Early Aptian black shales

Both, low TOC and high-TOC black shales contain thermally immature OC as indicated by the continuous down-core presence
of 5β-steranes and 17β,21β-hopanes. ββ-Hopane ratios do not increase downcore and show no systematic trend with TOC





(Figure 7b). Hence, changes in the sterane and hopane biomarker distribution due to thermal maturation are negligible and are

interpreted to reflect primary variations in the source and/or preservation of OC.

Early Aptian low-TOC background sediments contain a mixture of marine and terrestrial OC with a greater contribution of terrestrial OC compared to high-TOC black shale intervals, as indicated by (1) lower LMW/HMW $n$-alkane ratio of 1.3 (Figure 7c), (2) lower sterane/hopane ratios (Figure 7d), indicating a higher contribution from bacteria, typical for land-derived OC (e.g. Moldowan et al., 1985), (4) lower 24-npc/$C_{29}$-sterane ratios of ~0.13 (Figure 7e), and (3) higher abundances of $C_{29}$-sterane

(Figure 8). These results are consistent with previously published Rock-Eval data that show the dominance of terrigenous type III kerogen in low-TOC black shales (Hartwig et al., 2012) and microscopic inspection of organic matter that revealed the presence of abundant terrestrial ligneous debris (Raynaud and Robert, 1978).

Deposition of high-TOC black shales was accompanied by changing proportions of marine and terrestrial OC. High-TOC black shales mainly comprise marine OC, as indicated by the dominance of LMW $n$-alkanes (Figure 7c). In connection with

high sterane/hopane ratios (Figure 7d), the dominance of algal $C_{27}$-sterane and $C_{28}$-sterane over higher land plant-derived $C_{29}$-sterane support a predominantly marine OC source (Figure 8; Huang and Meinschein (1979); Moldowan et al. (1985)). Increased 24-npc/$C_{29}$-sterane ratios similarly suggest enhanced input of marine OC (Figure 7e), as 24-npc has been linked to OC inputs from chrysophyte algae (Moldowan et al., 1990) and/or Rhizaria (Nettersheim et al., 2019), a group of heterotrophic unicellular protists, including foraminifera and radiolaria, that occurs ubiquitously in the global ocean and plays a key in the

export of carbon from the photic zone to sediments (Caron, 2016; Lampitt et al., 2009). Based on above lines of evidence, we conclude that high-TOC black shales mark episodes of enhanced productivity and preservation of marine OC in the Cape Basin. This is further supported by the occurrence of II and IIS kerogens with hydrogen index values of up to 700 mg g$^{-1}$ TOC in high-TOC black shales (Herbin et al., 1987; Hartwig et al., 2012).

### 4.3 Depositional processes and provenance of Early Aptian black shales

Early Aptian black shales are intercalated with abundant sandstones, sandy mudstones, and siltstones (Figure 2a), which represent turbidites deposited in a fan to fan-valley environment (The Shipboard Scientific Party, 1978; Natland, 1978). This raises the question as to whether high-TOC/low-TOC black shale alternations reflect changes between turbiditic and hemipelagic sedimentation, similar to those observed at numerous sites in the opening North Atlantic and South Atlantic basins (Degens et al., 1986; Forster et al., 2008). At Site 361, all samples, however, exhibit Si/Al and Zr/Al ratios close to AS,

indicating a minor coarse siliciclastic component (Figure 2e,f). Similar Si/Al and Zr/Al ratios between high- and low-TOC black shales indicate that the proportion of coarse siliciclastics did not change in concert with TOC (Figure 3a), supporting that both black shale types are of hemipelagic origin.

Previous provenance studies at Site 361 indicate that the proximal S-African continent was a dominant sediment source to the Cape Basin during the Aptian (Natland, 1978). Similar to today, sediments accumulating in the deep Cape Basin were most

likely transported to the SW African shelf by west-flowing river systems, including the paleo-Karoo River, which entered the S-Atlantic close to the modern Olifants River mouth in proximity to Site 361 (Figure 3a; De Wit (1999)). Recent apatite fission track analyses in the coastal area of SW Africa date the incision of the paleo-Karoo River valley at ~120 to ~110 Ma (Kounov et al., 2008), implying active river input to the Aptian Cape Basin. Accordingly, we attribute variations in the geochemical composition of the sedimentary record at Site 361 to changes in sediment provenance and/or weathering regime on the S-

African continent. Consistent with paleo-drainage reconstructions (De Wit, 1999), we consider three potential sediment source regions, which cover the majority of the SW African continent (Figure 3a): (1) Paleozoic to Early Mesozoic sediments from the Karoo and Cape supergroup, mainly located in the SW and central part of S-Africa, (2) Jurassic volcanics related to the Karoo large igneous province, capping and intervening the Karoo supergroup in the E and NE, and (3) Archean to Proterozoic sedimentary, volcanic, and metamorphic rocks of the Kaapvaal Craton, located in the NE. The geochemical composition of

potential sediment source regions is presented in Figure 3c–e. Due to the complex geology of the Kaapvaal Craton and related





intracratonic sediment sequences, we also included geochemical data of river bed sediments from modern rivers draining the Kaapvaal Craton, reflecting a more integrated signal (Garzanti et al., 2014).

Trace element discrimination plots suggest that Paleozoic sediments from the Karoo and Cape supergroup were the dominant weathering source of low-TOC black shales (Figure 3c–e). This indicates a minor contribution from the Kaapvaal Craton and
Karoo volcanics in the E and NE, which display more mafic geochemical signatures, as indicated by lower Th/Sc, Zr/Sc, La/Sc, higher Ti/Zr (Figure 3c,d) and light REE depletion (i.e. REE characteristics of the Kaapvaal Craton are approximated by average Archean upper crust composition; Figure 3e). Hence, we propose a dominant sediment input from Paleozoic source regions located along the coast in the W and SW, with a minor contribution from mafic rocks further inland. This might indicate that sediment input predominantly originated from areas located W of the escarpment (Figure 3a), which today represents a
major drainage divide that possibly already existed during the Early Cretaceous (Moore et al., 2009).

In contrast, trace element and REE compositions of high-TOC black shales support a shift to more mafic signatures, typical for the Kaapvaal Craton and/or Karoo volcanics (Figure 3c–e), suggesting enhanced sediment supply from the N and NE (Figure 3a). These data indicate a stronger sediment supply from areas located in the continent's interior relative to more coastal areas. Based on this trend, we propose that the shift in provenance from low- to high-TOC black shales reflects
enhanced moisture supply to the continent's interior, augmenting river run-off in the upstream regions of the paleo-Karoo River and/or increasing chemical weathering intensity of basaltic rocks.

Furthermore, high-TOC black shales are characterized by overall high K/Al ratios of ~0.4 (Figure 2d), which fall close to the stoichiometry of illite (Figure 3b; Weaver and Pollard (1973)). Consistent with their more mafic trace element signature, the Kaapvaal Craton and Karoo volcanics show K/Al ratios substantially lower than those recorded in high-TOC black shales
(Figure 3b), indicating that higher K/Al ratios cannot be explained by the change in sediment provenance proposed for high-TOC black shales. Instead, we link K/Al ratios to a change in the weathering intensity of K-rich source regions along the coast (i.e. located west of the escarpment; Figure 3a), including argillaceous sediments of the Karoo and Cape supergroups, Cape Granites, and pan-African terranes of the Saldania Belt (Figure 3a). Today, rivers draining these felsic rocks supply erosional inputs particularly rich in illite, which accounts for up to 90% of the total clay mineral assemblage in surface sediments near
their river mouths (Birch, 1978). Quaternary sediment records also show that strong physical erosion and increased fluvial activity in these coastal source regions are reflected in enhanced input of K to the SW African shelf (Hahn et al., 2016), and probably the deep southern Cape Basin (Dickson et al., 2010). Assuming analogous mechanisms for the Cretaceous, we propose that the recorded increases in K/Al ratios at Site 361 reflect periods of strong physical erosion of coastal K-rich source regions and riverine discharge to the Cape Basin. Supporting evidence comes from the good agreement of K/Al ratios of high-
TOC black shales and unweathered Karoo sediments (Figure 3b), suggesting efficient bedrock erosion with minor alteration of the sediment's chemical composition by chemical weathering during transport from source to sink.

Based on above lines of evidence, we propose that high-TOC black shales were deposited during episodes of enhanced precipitation and strong river run-off from the proximal S-African continent, which caused enhanced contribution of weathering inputs from the continent's interior and intensification of bedrock erosion along the coast.

**4.4    Climate simulations**

Atmospheric processes including changes in precipitation, run-off, and wind-driven oceanic upwelling, directly or indirectly influence the magnitude of marine OC burial (Wagner et al., 2013 and references therein). In order to test feedbacks on OC burial in the young South Atlantic basin, we modeled atmospheric and oceanographic circulation changes during OAE 1a. We conducted two model runs implementing 600 ppm and 1200 ppm $pCO_2$, the latter of which falls within the lower range of
$pCO_2$ estimates for OAE 1a (Naafs et al., 2016).

The climate zonation generated by our model indicates that Site 361 and the southern African continent were located in a temperate humid climate belt, influenced by the austral westerlies (Figure 9a). This simulated climate zonation is consistent





with previous general circulation model experiments (Fluteau et al., 2007) and supported by the Early Cretaceous distribution of climate-sensitive deposits on the African continent, which shows the dominance of evaporites north of a paleolatitude of

~40° S and coal deposits south of ~40° S, reflecting overall arid and humid conditions, respectively (Figure 1; Boucot et al. (2013); Chumakov et al. (1995)). This distribution implies that the descending limb of the southern paleo-Hadley cell separating the subtropical arid climate belt and the mid-latitude temperate humid climate belt was located north of Site 361 (Boucot et al., 2013; Chumakov et al., 1995). In accordance with this pattern, the simulated zonal mean edge of the Southern Hemisphere Hadley Cell (defined as the latitude of the first zero crossing of the atmospheric mass stream function at the 500

hPa level) is located at around 31° S and 32° S for the 600 and 1200 ppm experiments, respectively. This position is consistent with sparse geochemical data from Early Cretaceous (Albian) black shale sections from DSDP Site 530 in the Angola Basin and DSDP Site 511 on the Falkland Plateau at paleo-latitudes of ~37° S and ~58° S, respectively (Wagner et al., 2013).

In contrast to the modern location along the coast of SW Africa, which is strongly influenced by the Benguela upwelling system, both of our simulations indicate that no substantial upwelling occurred along the SW African margin during the Aptian

(Figure 9b,c). This is a direct consequence of the more poleward position of the southern tip of Africa in the Early Cretaceous and the associated change in the dominant large-scale wind regime. While the present-day Benguela upwelling is driven by the prevailing southeasterly trade winds under the subtropical limp of the Hadley Cell, the more southern position of the African continent during the Aptian leads to the predominance of the mid-latitude westerlies. An eastern boundary upwelling system is, however, simulated further to the north in the Angola Basin, even though its strength would have been limited by

the dimensions of the small basin. The stratigraphic record of DSDP Site 530 from this region unfortunately does not reach back into the Aptian to test this model outcome. Based on these results, we conclude that ocean upwelling had a negligible impact on the magnitude of OC burial at Site 361.

Our simulations, however, indicate major alterations in continental hydrology in response to changes in atmospheric $p$CO$_2$ levels. At 600 ppm $p$CO$_2$, annual mean precipitation in S-Africa (i.e. in an area we consider representative for the catchment

area based on our sediment provenance assessment; Figure 10a,b) accounts to 3.0 mm d$^{-1}$, more than twice the average annual precipitation in modern S-Africa (~1.3 mm d$^{-1}$; Noble and Hemens (1978)). Precipitation is particularly strong along the SW coast of Africa (Figure 10a), implying that sediment production and mobilization predominantly occurred in coastal areas, which is consistent with provenance data from low-TOC black shales.

A doubling of atmospheric $p$CO$_2$ leads to overall much higher precipitation south of Site 361 but drier conditions in the

subtropical regions further north (Figure 10b). This is in line with an enhanced hydrological cycle as consequence of increased atmospheric $p$CO$_2$ and global mean temperature (Held and Soden, 2006). As southern Africa is influenced by both regional aridification and enhanced precipitation, annual mean rainfall changes over the proposed catchment area amount to only 1% between both simulations (Figure 10c). However, the doubling in $p$CO$_2$ leads to a more pronounced seasonal cycle in the local rainfall with reduced precipitation from November to May and an increase during the austral winter. This seasonal increase of

continental precipitation coincides with the simulated surface run-off peak in both simulations during June to August (Figure 10d). The combination of reduced winter evaporation with soil saturation leads to a disproportionally higher simulated surface run-off at 1200 ppm $p$CO$_2$. Surface runoff volume fluxes increase by 40% for the annual mean and even double during July to September. The reduced precipitation during austral summer does not influence surface runoff, but leads to a reduction in the simulated annual mean drainage (i.e. subsurface runoff) from 6150 m$^3$/s to 5685 m$^3$/s.

**4.5    Driving mechanisms of organic carbon burial**

In the Cape Basin, the occurrence of black shales (TOC >3%) is limited to a discrete time interval during the Early Aptian, followed by a gradual decrease in OC burial throughout the Aptian-Albian. A similar trend, though with a slightly different timing, has previously been observed at Site 364 located in the Angola Basin (Behrooz et al., 2018). These data consistently indicate that black shale occurrences in the emerging South Atlantic basins were amplified during early stages of basin

evolution (Pérez-Díaz and Eagles, 2017; Dummann et al., 2020), arguing that basin configuration had a first order control on OC burial in the South Atlantic basins.

Internal heterogeneities in OC quantity and composition of the black shales, reflected in alternations between high-TOC and low-TOC black shales, suggest that the magnitude of OC burial varied on time scales $\leq$~1 Myr. Our paleo-redox reconstruction, biomarker data, and sediment provenance assessment indicate that high-TOC black shales reflect episodes of enhanced

production and preservation of marine OC under anoxic–euxinic conditions, accompanied by changes in provenance and weathering state of sediments delivered from the proximal S-African continent. The latter are interpreted to reflect an expansion of the catchment area in response to enhanced precipitation in the continent's interior and intense bedrock erosion along the coast due to increased river run-off. TOC contents approach their highest values in concert with maxima in river run-off (i.e. highest K/Al), which implies a tight coupling of marine OC production and burial to continental hydrology via changes

in riverine nutrient supply. This mechanism is supported by our general circulation model data for OAE 1a, indicating that (1) no substantial ocean upwelling occurred along the SW African margin during the Aptian, and (2) continental hydrology, in particular surface run-off, was subject to major fluctuations in response to atmospheric $pCO_2$ changes, consistent with our proxy data.

Our simulations identify changes in $pCO_2$ as a primary forcing mechanism, which may well be applicable to the different

phases of OAE 1a, including an initial warming pulse (hyperthermal). Conditions conducive to enhanced burial of OC in the Cape Basin, however, were not limited to OAE 1a, but recurred, as evident from other high-TOC black shale intervals in older and younger stratigraphic intervals. Their timing, duration, and pacing cannot be constrained at Site 361 due to the incomplete stratigraphic coverage, limiting our capability to identify underlying forcing mechanisms by general circulation modeling. Comparable alternations in Cretaceous black shale sequences have, however, been reported from both the North and South

Atlantic throughout the Cretaceous, which have been attributed to geochemical expressions of large scale climate variations driven by orbital forcing (Beckmann et al., 2005; Behrooz et al., 2018; Hofmann and Wagner, 2011; Wagner et al., 2013; Kolonic et al., 2005; Herrle et al., 2003). According to our simulations, Site 361 was located close to the subtropical descending limb of the paleo-Hadley Cell, a setting marked by strong latitudinal climatic gradients (Figure 10a). Previous studies indicate that marine sediments deposited beneath the outer limb of the paleo-Hadley Cell often show particularly pronounced OC

cycles, which are linked to climate variations, induced by orbitally paced shifts of climate belts (Wagner et al., 2013). Hence, we speculate that recurrent OC burial maxima in the Cape Basin might be related to hydrological fluctuations on the South African continent in response to orbitally driven climate variability. This hypothesis may be tested in the future once new, high-resolution sediment records from the study area become available. Irrespective of the exact driving mechanisms, the consistency of trends in the proxy records in all high-TOC black shale intervals, regardless of their stratigraphic position,

supports that they were the result of similar forcing processes and likely triggered by enhanced riverine nutrient supply to the young South Atlantic.

### 4.6    Wider implications

Our study re-emphasizes that ocean basins emerging during the break up of super-continents provided favorable conditions for enhanced OC burial through basin geometries promoting hydrographic isolation and a close coupling of atmosphere–land–

ocean interactions. The results of this study further suggest that deep marine environments (i.e. >2000 m) along the continental margins of emerging ocean basins were sensitive to changes in nutrient fluxes from the continent. One underlying reason for this higher sensitivity to across-shelf nutrient transport may have been very narrow shelves, permitting substantial and almost direct escape of continent-derived nutrients and mineral matter to the deep ocean, which is supported by recent modeling data (Sharples et al., 2017; Izett and Fennel, 2018). Shelves in emerging ocean basins are generally narrow and underdeveloped, as

some time is required for drainage systems to mature and marginal strata to accumulate (Ravnås and Steel, 1998; Trabucho Alexandre et al., 2012). Consistent with these characteristics, turbiditic sandstones deposited at Site 361 lack re-deposited

benthic organisms, which has been interpreted to reflect short residence times or even no intermediate storage in near-shore or shelf environments.

Furthermore, our study provides constraints for the position of the descending limb of the southern Hadley Cell during the
Early Cretaceous, which had proven difficult based on previously available data and coring sites (Wagner et al., 2013).

## 5     Conclusions

In this study we reconstruct the evolution of paleo-redox conditions, OC composition, and sediment provenance in the Aptian–Albian Cape Basin based on Fe-S-TOC systematics, distribution of redox-sensitive trace metals, lipid biomarker data, and the inorganic geochemical composition of sediments at Site 361. The results demonstrate that tectonic restriction of the Cape Basin
during the Early Aptian generally promoted oxygen-deficiency in the deeper water column, favoring the deposition of extensive black shales. The richness and composition of OC in Early Aptian black shales at Site 361 varied on time scales <1 Myr, including the OAE 1a and several other events, documented in up to eight additional high-TOC (5–20%) black shale periods. These high-TOC events were the result of enhanced burial of marine OC deposited under anoxic, and probably euxinic bottom water conditions. Changes in OC burial were tightly coupled to fluctuations in river run-off from the southern African
continent, which, in turn, were triggered by aridity/humidity variations induced by shifts in the precipitation pattern in the temperate mid-latitude climate belt, as indicated by sediment provenance data. General circulation model data support this conclusion. Based on this tight coupling of biogeochemical cycling in the deep (>2000 m) Cape Basin and climate–land–ocean interactions, we propose that narrow immature continental shelves in emerging ocean basin facilitated a more efficient land–open ocean nutrient transfer. By Late Aptian–Albian times, the abatement of tectonic restriction of the Cape Basin finally
terminated the conditions favorable for OC preservation, as reflected in a two-step increase in bottom water oxygenation and concomitant decreases in OC burial. This leads us to conclude that climatic fluctuations alone without the appropriate basin configuration for OC preservation were insufficient to generate enhanced OC burial.

## 6     Data availability

All original data of this publication will be made available through PANGAEA® Data Publisher for Earth and Environmental
Science (https://www.pangaea.de/).

## 7     Author contributions

PH, SF, JOH, JR, and TW were involved in the conceptualization of this study and acquired funding. WD and PH conducted XRF analyses. Biomarker analysis were conducted by WD under supervision of SK, CH, and JR. ICP-MS data were acquired by ML under supervision of HUK. SS and SF carried out the modeling experiments. WD integrated the data and wrote the
manuscript with contributions from all authors.

## 8     Competing interests

The authors declare that they have no conflict of interest.

## 9     Acknowledgments

We thank Nicole Mantke and Simon Kalinowski for their support in proxy data acquisition and Tamara Mai for her help during
sampling of Site 361. Model integrations were conducted at the Computing Center of Kiel University. We thank Janine




Blöhdorn for generating most of the Cretaceous KCM boundary conditions and Stefan Hagemann for constructing the parameters for the hydrological discharge model. This research used samples provided by the Deep Sea Drilling Project. PH, JOH and SF received funding for this work from the German Research Foundation DFG (grant numbers HO2188/9, HE3521/6).

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





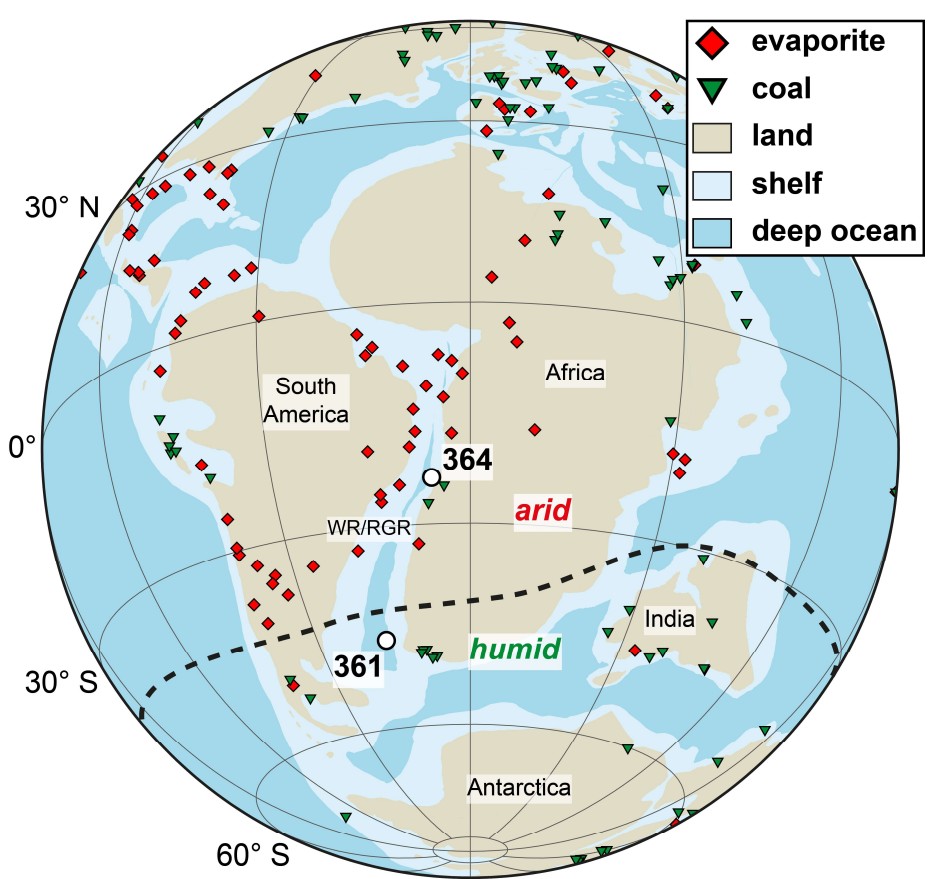

**Figure 1: Aptian paleogeographic reconstruction (Matthews et al., 2016) showing the Early Cretaceous distribution of climate**
**sensitive rock types (Boucot et al., 2013; Cao et al., 2019). Position of climate belts according to Boucot et al. (2013) and Chumakov et al. (1995). Land–sea mask and distribution of shelf seas were adapted from Cao et al. (2017). WR/RGR: Walvis Ridge/Rio Grande Rise.**



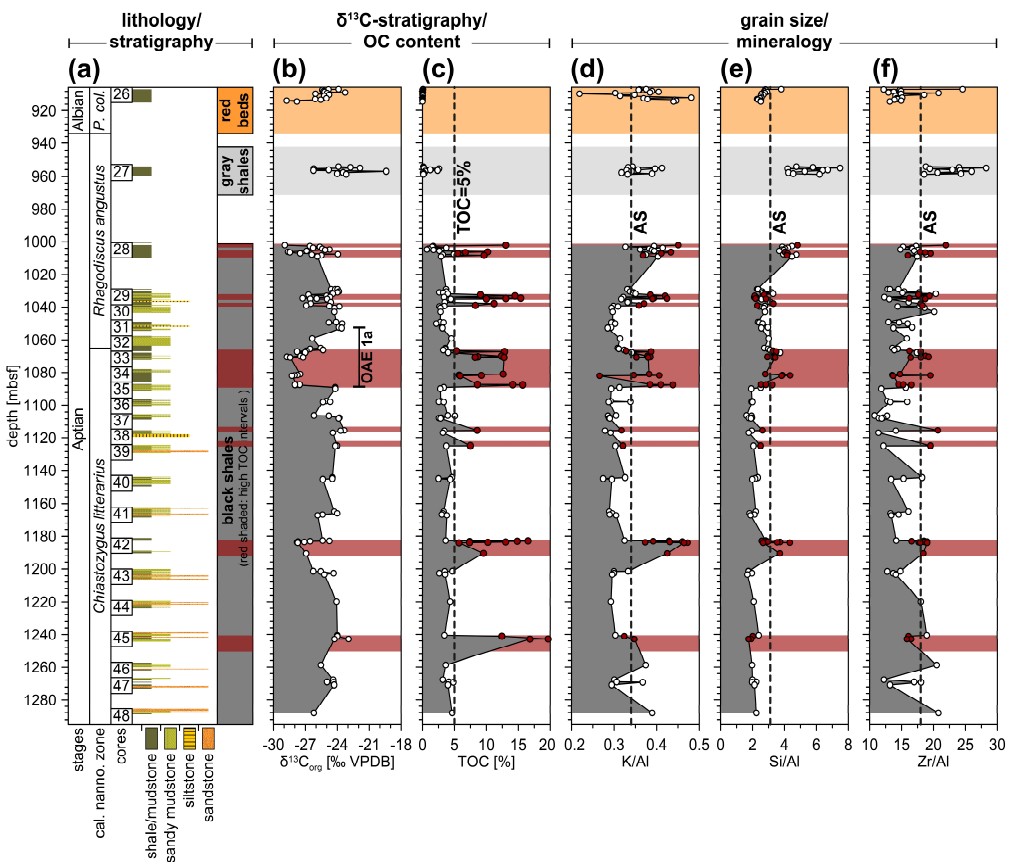

**Figure 2: (a) stage assignment, calcareous nannofossil zonation, and lithology of the studied sediment sequence at DSDP Site 361,**
**(b) bulk δ¹³C_org, (c) total organic carbon content, (d) K/Al ratios, (e) Si/Al ratios, and (f) Zr/Al ratios. Red background shading**
**represents stratigraphic position of main high-TOC black shale intervals. Dashed lines in (d)–(f) represent average shale (AS) ratios.**
**Bulk δ¹³C_org data and TOC content are taken from Dummann et al. (2020).** *P. col. = Prediscosphaera columnata*


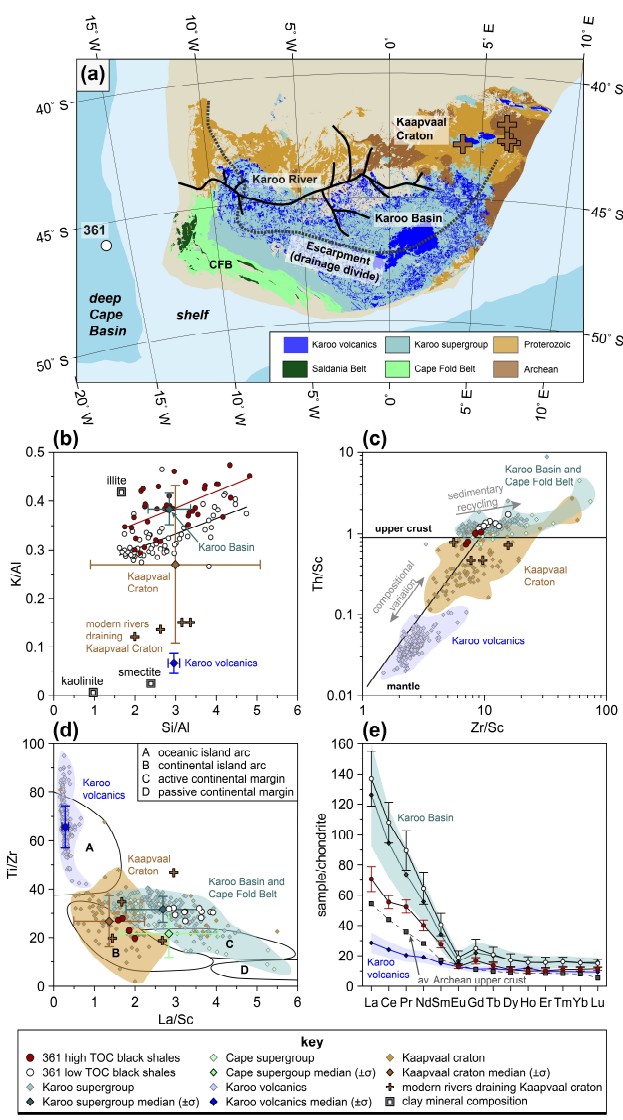

**Figure 3: Provenance assessment of Early Aptian black shales based on major and trace metal discrimination plots: (a) geological map of South Africa showing potential sediment source areas. Early Cretaceous paleo-drainage reconstruction was adapted from De Wit (1999). (b) Si/Al vs. K/Al plot. Average elemental composition of different clay minerals was taken from Weaver and Pollard (1973). (c) Zr/Sc vs. Th/Sc (McLennan et al., 1993), (d) La/Sc vs. Ti/Zr (Bhatia and Crook, 1986), and (e) chondrite-normalized rare earth element distribution. Trace element data for source areas were compiled from the literature. Data and references are provided in the Supplement S2 and S3.**





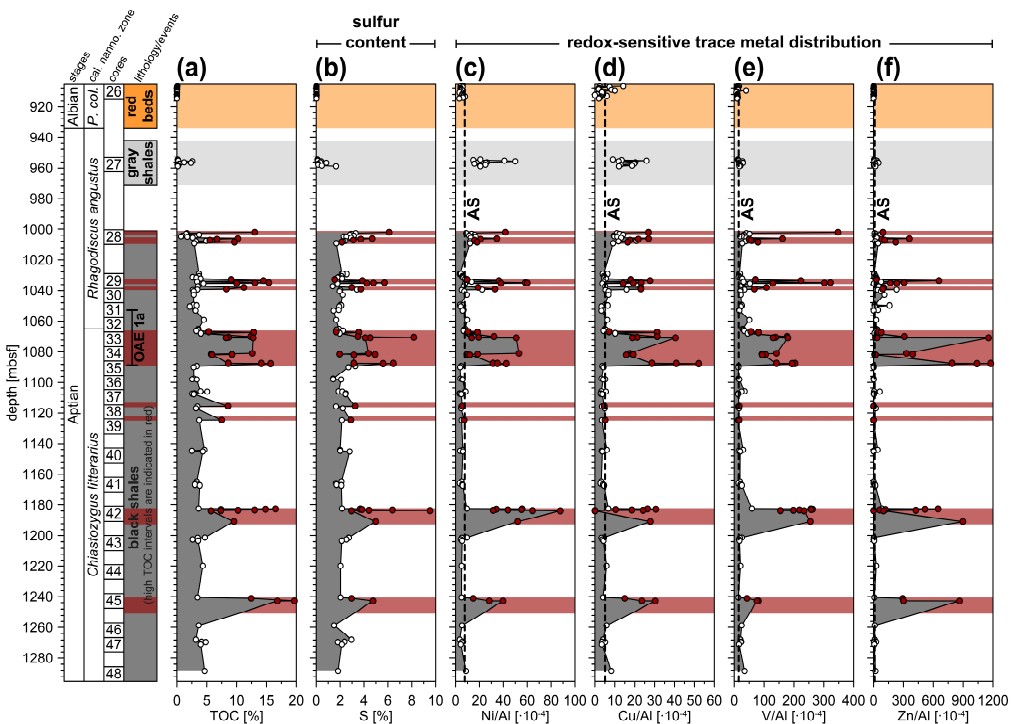

**Figure 4: Inorganic geochemical parameters used to reconstruct paleo-redox conditions: (a) total organic carbon content (Dummann et al., 2020), (b) total sulfur content, (c)–(f): Ni/Al, Cu/Al, V/Al, and Zn/Al ratios. Average shale TM/Al ratios (AS) are plotted as dashed lines. *P. col. = Prediscosphaera columnata***






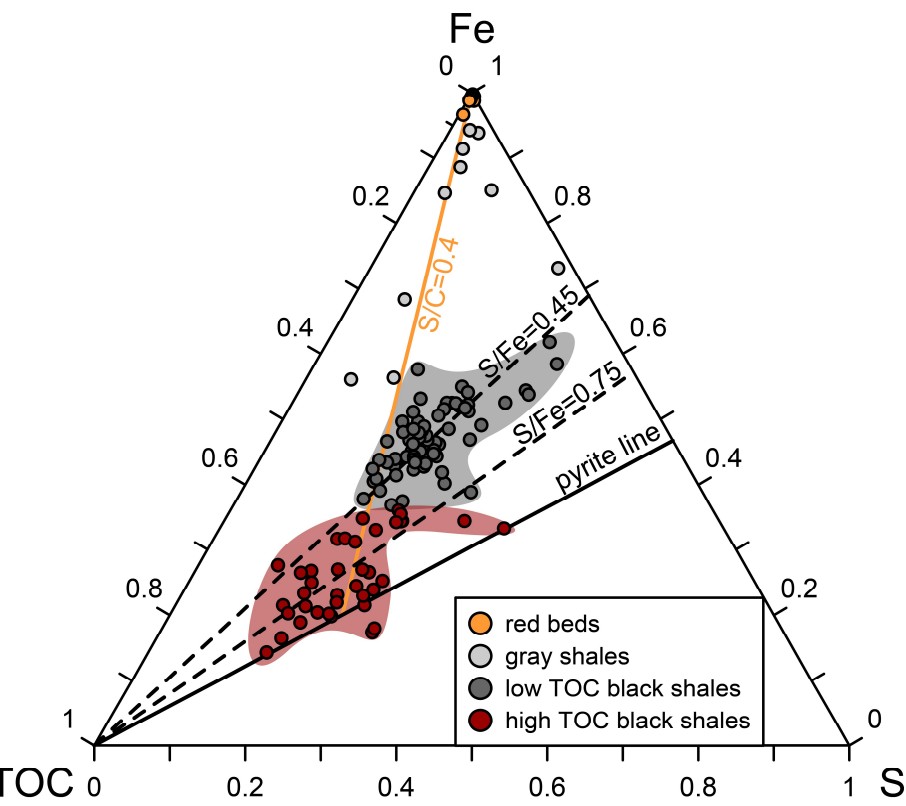

**Figure 5:** Ternary diagram illustrating TOC-S-Fe relationships of red bed, gray shale, low-TOC black shale (<5% TOC), and high-TOC black shale (>5% TOC) samples. Characteristic S/Fe and S/C ratios are plotted as proposed by Raiswell et al. (1988), Leventhal (1983) and Berner and Raiswell (1983). TOC data are taken from Dummann et al. (2020).



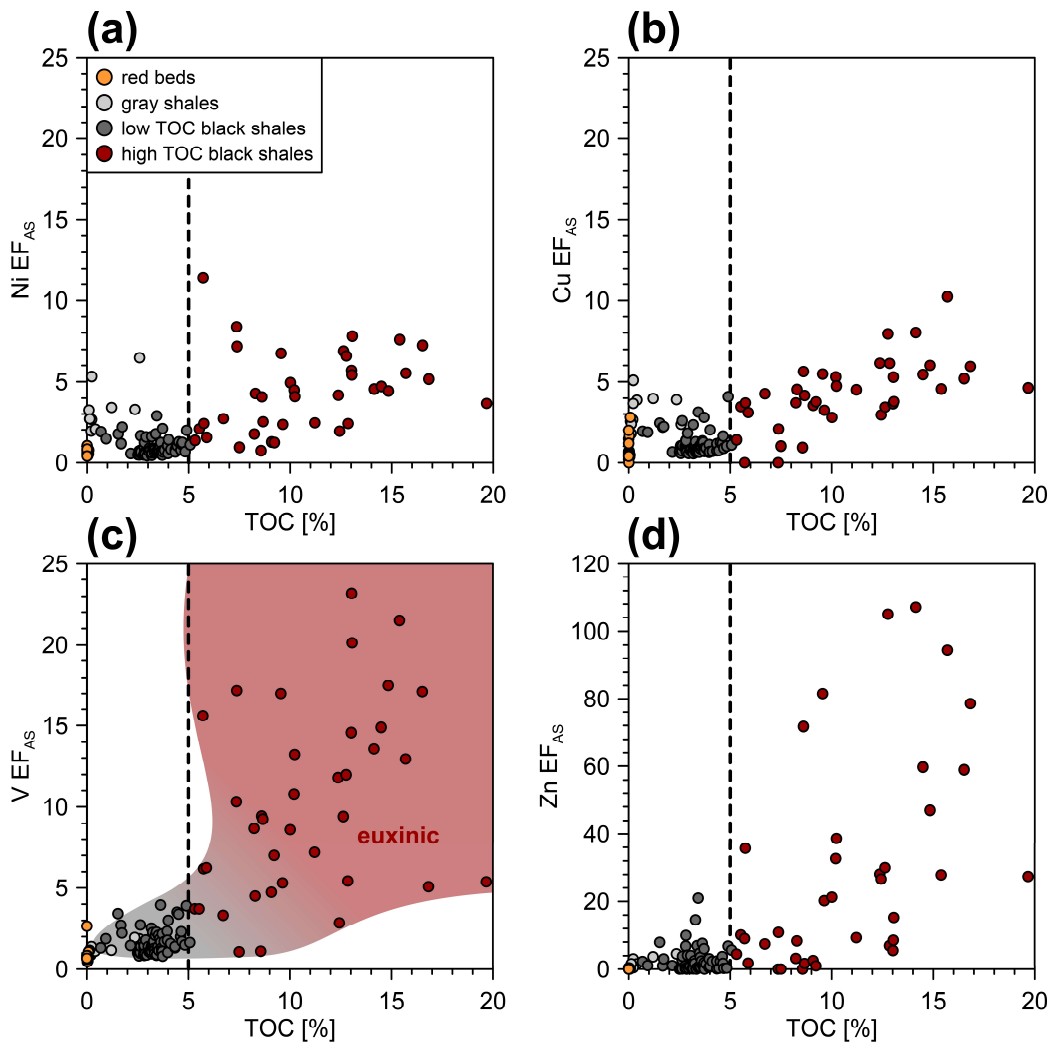

**Figure 6:** Cross-plots of total organic carbon content and average shale (AS)-normalized enrichment factors of (a) Ni, (b) Cu, (c) V, (d) Zn. Key is given in (a). Low-TOC black shales are defined as <5% TOC, while high-TOC black shales contain >5% TOC. TOC data are taken from Dummann et al. (2020).






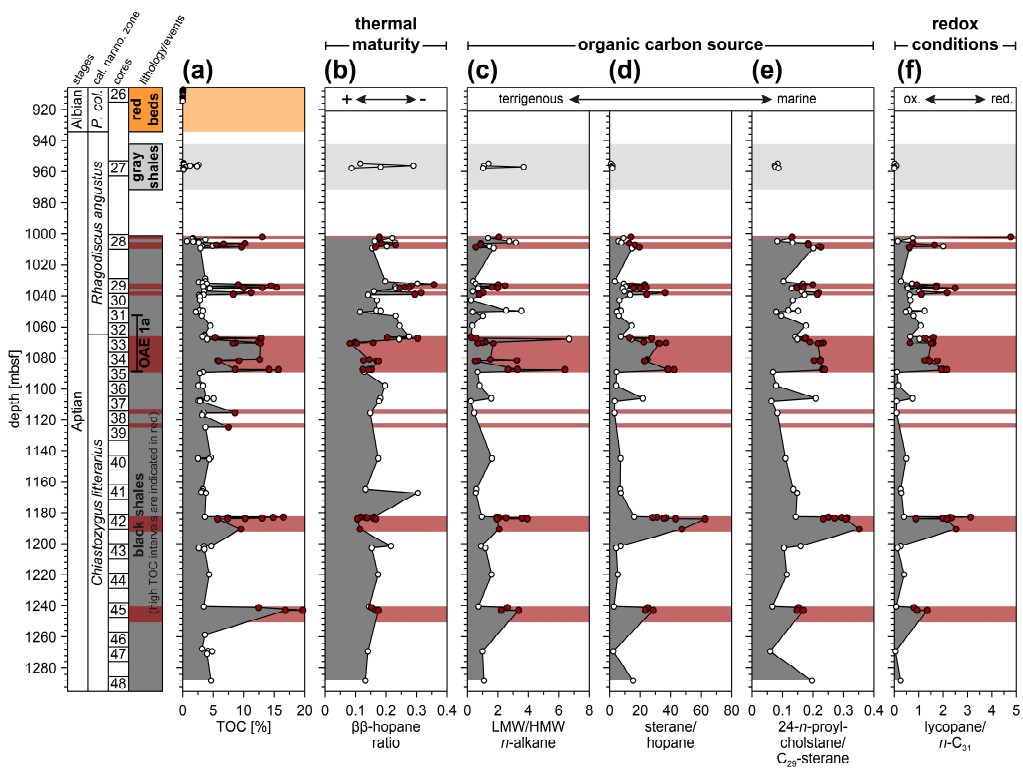

**Figure 7:** **Distribution of selected biomarker parameters used to reconstruct thermal maturity, OC source, and paleo-redox conditions: (a) total organic carbon content (Dummann et al., 2020), (b) $\beta\beta/(\beta\beta+\alpha\beta+\beta\alpha)$-$C_{30}$-hopane ratio, (c) low molecular weight/high molecular weight $n$-alkane ratio, (d) sterane/hopane ratio, (e) 24-$n$-propylcholestane/$C_{29}$-sterane ratio, and (f) lycopane/$n$-$C_{31}$ ratio.** *P. col.* = *Prediscosphaera columnata*



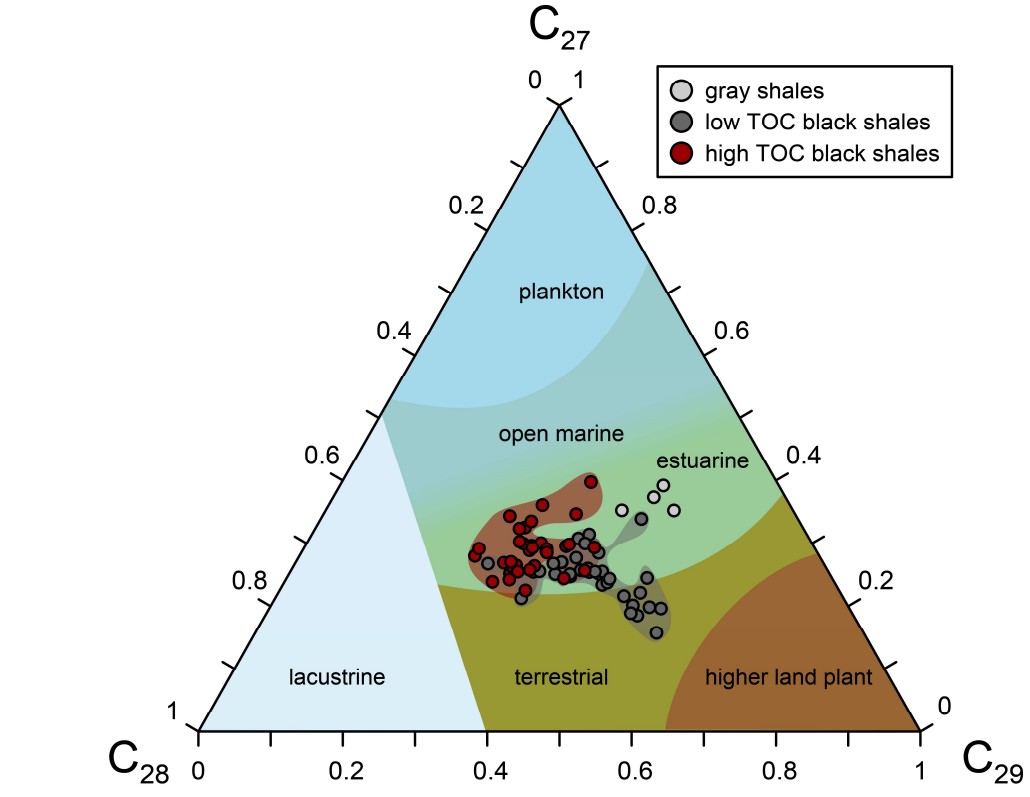


**Figure 8: Ternary diagram showing the distribution of regular desmethylsteranes in gray shales, low-TOC black shales (<5% TOC), and high-TOC black shales (>5% TOC). Organic carbon source endmembers were adapted from Huang and Meinschein (1979).**



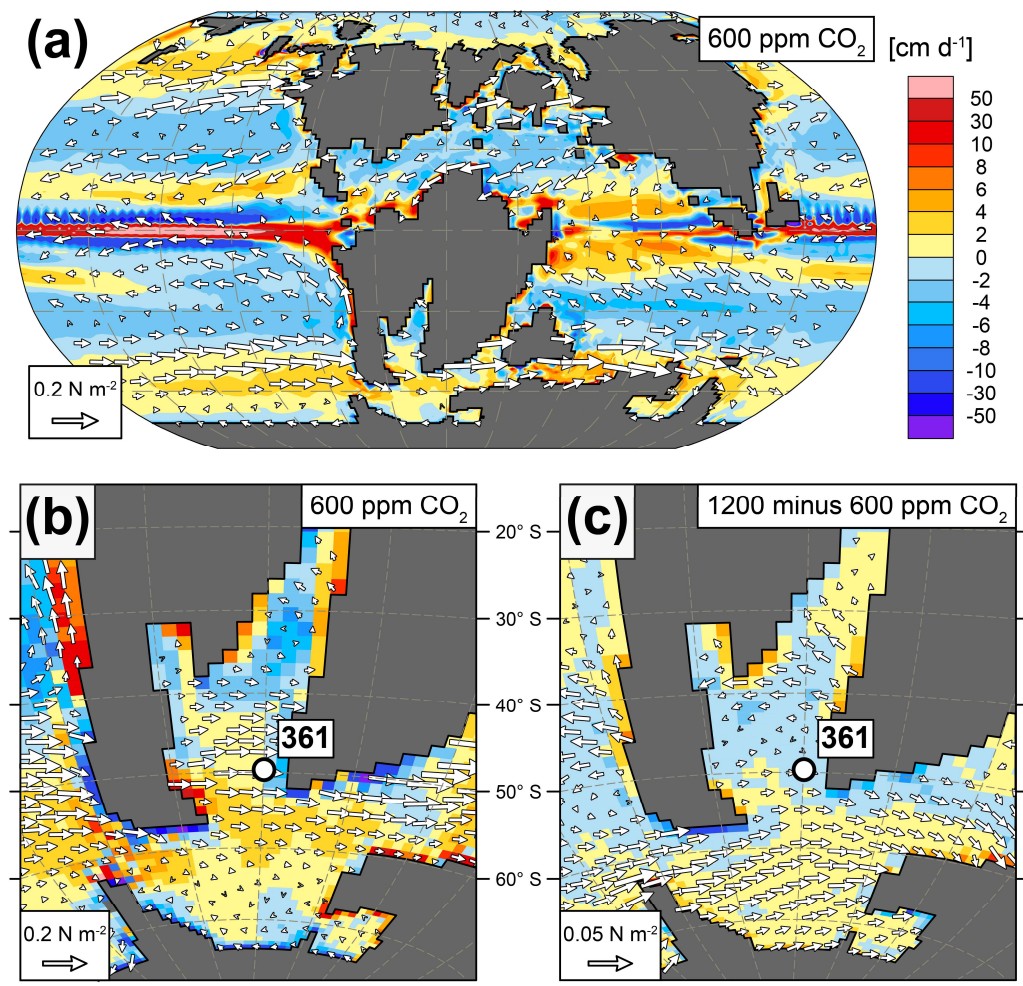

**Figure 9: (a) Annual mean surface wind stress and ocean vertical velocity at 10 m water depth simulated by the KCM at 600 ppm**
$p$CO₂. **Positive contours indicate upwelling. (b) same as in (a) but for the S-Atlantic basins as well as (c) the relative change due to a doubling of atmospheric** $p$CO₂**.**





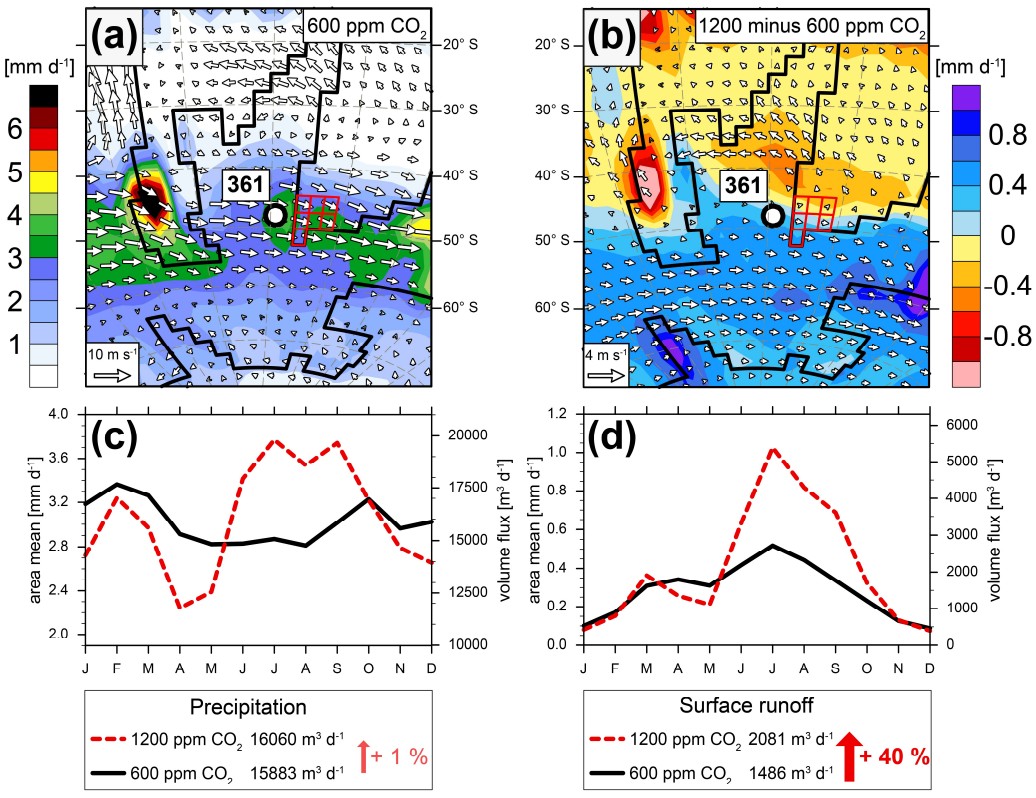

**Figure 10: (a) Simulated annual mean surface wind and total precipitation in the study area at 600 ppm $p$CO₂, (b) relative surface**
**wind and precipitation change due to a doubling of atmospheric $p$CO₂. (c) Monthly mean precipitation and (d) generated surface runoff averaged over the catchment area indicated by the highlighted grid points in (a) and (b). Volume fluxes are integrated over the whole catchment area (~450.000 km²) with annual mean values reported in the respective legends.**