# Peer review of "Driving mechanisms of organic carbon burial in the Early Cretaceous South Atlantic Cape Basin (DSDP Site 361)"

_Climate of the Past, 2020_

## Referee Comment (RC1) · Alexandre Pohl (Referee) · 9 Mar 2020

**Paper summary**:

Dummann et al. investigated the mechanisms driving organic carbon burial during the Aptian–Albian in the deep Cape Basin at Deep Sea Drilling Project (DSDP) Site 361. To that purpose, the authors produced abundant (bulk and molecular) geochemical data and conducted general circulation model runs using the ocean-atmosphere Kiel Climate Model. Dummann et al. identified two main temporal trends based on the sedimentary record: (1) a long-term trend toward more oxygenated conditions, as can be seen based on the changes in sedimentary facies from black shales to grey shales and red beds, and (2) a higher-frequency change, inside the Aptian black shale unit, between low-TOC black shales and high-TOC black shales (the latter comprising OAE1a, but also other layers). The authors explain the reported long-term change in deep-water redox conditions at DSDP Site 361 by a reorganization of the ocean circulation in response to the evolution of marine gateways, based on the findings of their previous study (Dummann et al., 2020). Regarding the high-frequency cycles, proxy data demonstrate a common geochemical signature of the high-TOC black shales, suggesting that the same mechanisms potentially drove the deposition of the successive layers. These data point toward an increasing contribution of marine organic matter during the deposition of these high-TOC shales. The systematic increase in the K/Al ratio during these periods suggests that the latter may correspond to periods of increased precipitation and runoff from the continent, bringing unweathered sediments and also nutrients to the ocean, thus explaining the increase in marine primary productivity and organic matter deposition under anoxic–euxinic conditions. In order to check this hypothesis, the authors conducted 2 climatic simulations at 600 ppm and 1200 ppm $CO_2$, supposedly representative of Aptian background climate and OAE1a. Although total precipitation over the region of interest does not significantly increase in response to climate warming in the model, spatial contrasts strengthen and runoff during the austral winter doubles. From this point of view, simulations support the authors' hypothesis.

**General comment**:

As a paleoclimate modeler interested in proxy data but not expert in geochemistry, I really appreciated the manuscript. Geochemical analyses and results are exposed in a very clear and accessible way. The authors further use a climate model to validate the hypotheses established based on the analysis of the geological record. The manuscript is well organized and the statements are mostly very well supported by analyses/figures of high quality. I recommend rapid publication of this study after minor revisions.

**Major comments**:

**1**. Model-data comparison and the change in sediment supply between low-TOC and high-TOC shales. On lines 414–416, the authors "*propose that the shift in provenance from low- to high-TOC black shales reflects enhanced moisture supply to the continent's interior, augmenting river run-off in the upstream regions of the paleo-Karoo River and/or increasing chemical weathering intensity of basaltic rocks*". However, results of the climate model suggest that coastal S and SW coastal regions became more humid during OAE1a-like warming events, while the continent interior became more arid. These two elements – the hypothesis derived from proxy data and the results of the GCM – thus seem in contradiction. This discrepancy should be resolved.

**2**. The paleo-Karoo River. The existence of this river has a major role in the hypotheses drawn in this contribution. Are the arguments supporting (1) the existence of the river and (2) the location of the river mouth during the Aptian (especially lower Aptian) robust? Topography, in particular, is generally very difficult to reconstruct in deep-time periods, thus making the reconstruction of watersheds challenging, too. I suggest expanding the arguments supporting the existence of this river, of its watershed such as drawn in Fig. 3 and of the location of the river mouth. If arguments are solid,

such discussion will strengthen the manuscript. If they are not, I still believe that the authors' arguments are very interesting but I think it would be good in that case to highlight these uncertainties as a limitation.

**3**. GCM simulation and OAE1a. The authors suggest that a pCO2 doubling from 600 ppm to 1200 ppm is well representative of the climate change that accompanied OAE1a. The choice of the pCO2 values is based on the argument that these values fall "*within the lower range of pCO2 estimates for OAE 1a (Naafs et al., 2016)*" (also lines 160–163). However, different climate models have different climatic sensitivities, i.e., a given change in pCO2 will not have the same effect depending on the climate model in use. Is the climate change simulated in the Kiel model between 600 and 1200 pm really representative of what we know of OAE1a? Interestingly, recent temperature estimates (Naafs and Pancost, 2016; O'Brien et al., 2017) provide an overview of the ocean warming that accompanied OAE1a. I think it would be instructive to provide a rapid comparison between the modeled temperatures and these recent proxy data. Such comparison would satisfactorily complement the interesting comparison of the modeled climatic belts with the lithological indicators of climate already proposed by the authors.

**Minor comments**:

**1**. Lines 441–442: the authors state that "*The climate zonation generated by our model indicates that Site 361 and the southern African continent were located in a temperate humid climate belt*". How was this climatic belt identified? Was is through some Köppen classification or similar? Please support this statement.

**2**. Lines 444–446: I suggest including a reference to the modeling study of Chaboureau et al. (2012), which specifically investigated the controlling mechanisms for the deposition of evaporites in the northern and central sections of the nascent South Atlantic Ocean.

**3**. Lines 464–466: I suggest comparing the precipitation simulated at 600 ppm with the precipitation simulated in modern S-Africa in the present-day model control run. This might be more instructive (consistent) than comparing to an independent dataset.

**4**. Lines 496: "*no substantial ocean upwelling occurred along the SW African margin during the Aptian*". Here as well, I suggest approaching the question of the limitations. Indeed, the authors are here reaching the limit of the modeling setup. Coastal upwellings s.s. are 10s of km wide, not 100s. As a consequence, they cannot be captured by the ocean model, the mean horizontal resolution of which is ca. 200 km.

**5**. Figs. 9–10: Since the position of Site 361 is critical, please state how its paleocoordinates were calculated and how the point was subsequently located on the mixed Blakey-Sewall map.

**Technical comments**:

- Line 129: "*The distribution of acyclic hydrocarbons (i.e. n-alkanes, acyclic isoprenoids)  was analyzed*"
- Line 207: "*Distribution of the redox-sensitive trace metals (TMs) Ni, Cu, V, and Zn in black shales  is presented as*"
- Line 417: "*furthermore*" is misleading because the new idea is not used to support the previous paragraph.
- Fig. 2 caption: Missing general title for the Fig.
- Fig. 9 caption: "Positive contours" > positives values (there are no contours).

**References cited**:

Chaboureau, A.C., Donnadieu, Y., Sepulchre, P., Robin, C., Guillocheau, F., and Rohais, S., 2012, The

Aptian evaporites of the South Atlantic: A climatic paradox? Climate of the Past, v. 8, p. 1047–1058, doi:10.5194/cp-8-1047-2012.

Dummann, W., Steinig, S., Hofmann, P., Flögel, S., Osborne, A.H., Frank, M., Herrle, J.O., Bretschneider, L., Sheward, R.M., and Wagner, T., 2020, The impact of Early Cretaceous gateway evolution on ocean circulation and organic carbon burial in the emerging South Atlantic and Southern Ocean basins: Earth and Planetary Science Letters, v. 530, p. 115890, doi:10.1016/j.epsl.2019.115890.

Naafs, B.D.A., and Pancost, R.D., 2016, Sea-surface temperature evolution across Aptian Oceanic Anoxic Event 1a: Geology, v. 44, p. 959–962, http://geology.gsapubs.org/lookup/doi/10.1130/G38575.1.

O'Brien, C.L. et al., 2017, Cretaceous sea-surface temperature evolution: Constraints from TEX 86 and planktonic foraminiferal oxygen isotopes: Earth-Science Reviews, v. 172, p. 224–247, doi:10.1016/j.earscirev.2017.07.012.

---

## Referee Comment (RC2) · Anonymous Referee #2 · 29 Jun 2020

This manuscript argues that variation between high TOC and modest TOC Aptian black shales at DSDP 361 in the Cape Basin was forced by changes in precipitation patterns over southern Africa. These relatively short-term climatic variations are proposed to be separate from a longer-term tectonic shifts that led organic carbon-rich deposits to be replaced organic carbon-poor deposits from the Aptian through the Albian at this site. Primary evidence cited in support of the climatic interpretation is biomarker analyses, REE distributions, and modelling results. In addition, elemental abundances and other bulk geochemical results are discussed relative to potential diagenetic overprints and redox conditions in the water and sediment column. Data are largely limited to a portion of the lithologies present at one site that seemingly had quite incomplete recovery.

[Figure]

The interpretation is interesting and consistent with the data presented, but to fit the observations, the interpretation does invoke a rather specific set of conditions (e.g., changes in seasonality and geographic focus of precipitation leading to more precipitation and nutrient input from chemical weathering in the interior but more physical weathering along the coast) that are difficult to test. The study also leans heavily on Dummann et al., 2020, EPSL, for data and documentation to the point that I wonder if the last sentence of the first paragraph of the discussion of the EPSL paper refers to the data split off and presented in this manuscript. Regardless of decisions made by authors about whether and how to partition findings among publications, this manuscript would be stronger if the unique contributions of this study were featured and less prominence was given to the broader and longer-term topics addressed in the EPSL paper.

Emphasizing and expanding presentation of sedimentological data would be one route to this end especially given the importance of deposition patterns and sediment sources in the conclusions. Currently the lithologic data presented are relatively sparse. The paper proposes 3 (or 4) lithostratigraphic units are present that can be differentiated based on color and TOC of the finest grained lithologies. Is there any information on mineralogy, bedding characteristics, or sedimentary structures (physical or trace fossils)? Description of the samples analyzed, their context, and why they were chosen is effectively missing (both for the 'complete set' and the presumably subsets thereof in plots with < than 131 data points). What about the coarser grained lithologies? It seems these lithologies are an important part of the depositional system, but they are mentioned mostly for having been excluded from consideration. The graphic column, on the other hand, seems to show changes in the proportion of lithologies through time (as well as substantial coring gaps). How does the stratigraphic pattern of different lithologies fit into the provenance/sediment routing interpretation proposed? Petrographic and geochemical examination of the coarser lithologies could provide information about provenance and transport. Do K/Al and Si/Al measures of sedimentologically identified distal turbidites support the assertion that hemipelagic deposition

and turbidites can be separated by the metrics used at this site. How was the 5% organic carbon threshold chosen? Are there really two different lithologies (<5% and >5% Corg) within fine-grained portions of the "black shales" or is there a continuum? What is the stratigraphic character of the most TOC-rich intervals– discrete beds (how thick?) or trends over some thickness (cm? dm? m?). One way to illustrate/test this classification would be a plot of K/Al vs. %organic carbon (cf. Fig. 3b). Statistical tests of apparent geochemical patterns would enhance confidence in the relationships proposed. Were there efforts to separate lithogenic from hydrogenic contributions to the REE profiles measured? Could the differences in 3c-d between bulk analyses reflect oceanographic/diagenetic conditions correlated with organic carbon content rather than changes in sediment provenance causing changing in oceanographic/depositional conditions? Some of this information is in the EPSL manuscript and I assume some is in the DSDP volumes cited, but this study would be stronger if more supporting sedimentological and stratigraphic information was summarized and synthesized in the manuscript.

Along similar lines, the presentation of the modelling and modelling results is limited although those results arguably provide the only independent assessment of the model proposed. Description of the model (2.4) is brief and and the changes in bathymetry between the Dummann et al. 2020 EPSL paper's and the bathymetry used in this paper is not illustrated. The paleogeographic figures in the EPSL paper are superior to the rather simple Fig. 1 in this manuscript. In figures 9 and 10 it would be nice to see a panel that has the 1200 ppm CO2 results shown rather than simply showing 600 ppm results (9b, 10a) and a difference (9c, 10b). The idea that an intermediate water mass isolates the deep Cape Basin promoting organic carbon accumulation is an interesting model-based suggestion. Could cross sections across the basin showing model predicted water properties that indication a saline layer divides the water column be provided as well as sensitivity tests of the persistence and extent of such a layer under the two climatic modes proposed to explain the distribution of TOC in the black shales? Could discussion and supporting graphics demonstrating the models have

stabilized be provided especially relative to the deep basin?

Finally, the manuscript suggests that increased CO2 might be the cause of the proposed changes in precipitation leading to nutrient input and enhance physical weathering proposed to explain organic carbon enrichment associated with OAE1a. The manuscript also suggests orbital variations might be the cause of the other organic carbon rich black shale intervals. Are the authors proposing that orbital variations cause a change in CO2? If so, how? If they instead are suggesting both orbital variations and CO2 increase cause the same climatic changes, this would be an idea that seems worth including in section 4.6. Would the authors hypothesize what orbital configuration would be expected to mimic high CO2 precipitation patterns and why? Such a prediction could be tested when/if an orbital timescale is generated for the Aptian Cape Basin.

55- does the mid-ocean ridge represent a barrier to circulation? 57- how is global excess OC burial defined/calculated 59- Angola Basin not shown in Figure 63- which data- TOC percentages? 87- beyond TOC (and color?) are there sedimentological differences among lithostratigraphic divisions, gray shales and low-TOC black shales do not seem different based on the criteria used 95- describe sample set 126, 235- don't start sentence with lowercase letter, write around 144- not sure this is a method, odd to exclude sandstones for provenance work 149- These data 177- very different precision suggested for characteristic AS Si/Al and Zr/Al ratios 180- difference seems small given variance- is it statistically supported. What does a plot of TOC vs. K/Al look like? 254- is proposed difference statistically supported?

Figure 2- using red shading for organic carbon rich black shales and orange shading for red beds seems like an odd choice; panels d-f are labeled grain size/mineralogy, but the plots show elemental ratios.

Supplementary information could include some description of the plots.

---

## Author Comment (AC1) · 24 Aug 2020

**Author's response to referee comment (RC) 1**

**We would like to thank referee #1 (Alexandre Pohl) for his time to review our manuscript and for raising important questions that will help to improve the overall quality of our manuscript. We will address the main concerns below in a detailed point-to-point response. New Supplementary Figures 1-3 are referenced in this response and are included at the end of this document.**

Paper summary:
Dummann et al. investigated the mechanisms driving organic carbon burial during the Aptian–Albian in the deep Cape Basin at Deep Sea Drilling Project (DSDP) Site 361. To that purpose, the authors produced abundant (bulk and molecular) geochemical data and conducted general circulation model runs using the ocean-atmosphere Kiel Climate Model. Dummann et al. identified two main temporal trends based on the sedimentary record: (1) a long-term trend toward more oxygenated conditions, as can be seen based on the changes in sedimentary facies from black shales to grey shales and red beds, and (2) a higher-frequency change, inside the Aptian black shale unit, between low-TOC black shales and high-TOC black shales (the latter comprising OAE1a, but also other layers). The authors explain the reported long-term change in deep-water redox conditions at DSDP Site 361 by a reorganization of the ocean circulation in response to the evolution of marine gateways, based on the findings of their previous study (Dummann et al., 2020). Regarding the high-frequency cycles, proxy data demonstrate a common geochemical signature of the high-TOC black shales, suggesting that the same mechanisms potentially drove the deposition of the successive layers. These data point toward an increasing contribution of marine organic matter during the deposition of these high-TOC shales. The systematic increase in the K/Al ratio during these periods suggests that the latter may correspond to periods of increased precipitation and runoff from the continent, bringing unweathered sediments and also nutrients to the ocean, thus explaining the increase in marine primary productivity and organic matter deposition under anoxic–euxinic conditions. In order to check this hypothesis, the authors conducted 2 climatic simulations at 600 ppm and 1200 ppm CO2, supposedly representative of Aptian background climate and OAE1a. Although total precipitation over the region of interest does not significantly increase in response to climate warming in the model, spatial contrasts strengthen and runoff during the austral winter doubles. From this point of view, simulations support the authors' hypothesis.

**This summary is mostly correct, but we want to highlight that our manuscript actually covers three different driving mechanisms of organic carbon burial in the Cape Basin. We want to briefly reiterate these processes to avoid any potential misunderstanding and to facilitate our response to the remaining reviewer comments:**

(1) **Long-term (>1 Myr) tectonic changes gradually increased the South Atlantic-Southern Ocean deep water mass exchange, which in turn enhanced bottom water oxygenation and reduced organic carbon burial in the deep Cape Basin.**

(2) **Higher atmospheric $p$CO$_2$ during OAE 1a enhanced seasonal precipitation and surface runoff from the African continent. The associated increase in weathering-derived nutrient input into the South Atlantic temporarily promoted local organic carbon burial. This data-derived hypothesis is supported by our two high and low $p$CO$_2$ modelling experiments.**

(3) **In contrast to OAE 1a, we have no evidence that other recurring carbon burial maxima during the Aptian were linked to global $p$CO$_2$ changes. We rather propose that these short-term fluctuations were caused by orbitally driven changes in the seasonal inland precipitation over South Africa that ultimately controlled riverine nutrient input and productivity in the Cape Basin.**

General comment:

As a paleoclimate modeler interested in proxy data but not expert in geochemistry, I really appreciated the manuscript. Geochemical analyses and results are exposed in a very clear and accessible way. The authors further use a climate model to validate the hypotheses established based on the analysis of the geological record. The manuscript is well organized and the statements are mostly very well supported by analyses/figures of high quality. I recommend rapid publication of this study after minor revisions.

**We thank the reviewer for the positive feedback.**

Major comments:
**1**. Model-data comparison and the change in sediment supply between low-TOC and high-TOC shales. On lines 414–416, the authors "*propose that the shift in provenance from low- to high-TOC black shales reflects enhanced moisture supply to the continent's interior, augmenting river run-off in the upstream regions of the paleo-Karoo River and/or increasing chemical weathering intensity of basaltic rocks*". However, results of the climate model suggest that coastal S and SW coastal regions became more humid during OAE1a-like warming events, while the continent interior became more arid. These two elements – the hypothesis derived from proxy data and the results of the GCM – thus seem in contradiction. This discrepancy should be resolved.

**We argue that there is no discrepancy between the proxy-derived hypothesis and the modeling results but agree that the current presentation of the precipitation maps in Figure 10 can be improved. While the current annual mean map in Figure 10b indeed shows a more arid continental interior, our results reveal that it is actually the enhanced seasonality of precipitation over South Africa in the high $CO_2$ simulation (Figure 10c) that drives the overall increase in surface runoff (Figure 10d). Higher rainfall during austral winter (June to August) disproportionately increases surface run-off due to reduced evaporation and high soil moisture saturation. To better illustrate these seasonal fluctuations, we therefore decided to change the maps in Figure 10a-b and rather show averages for the months June to August instead of annual mean fields. The updated figure is attached as Supplementary Figure 1 and clearly shows that the increase in austral winter precipitation in response to a $CO_2$ doubling occurred over large areas in the South African hinterland. We conclude that these modeling results are in overall agreement with the data-derived hypothesis of enhanced river run-off during OAE 1a and should now be clearer to the reader. We will further emphasize the importance of the seasonal precipitation cycle for the surface run-off in the main text. Additionally, in response to a comment by reviewer #2, we will add seasonal and annual mean maps for both simulations to Supplementary Figure 2, to allow a more in-depth assessment of the modelling results.**

**2**. The paleo-Karoo River. The existence of this river has a major role in the hypotheses drawn in this contribution. Are the arguments supporting (1) the existence of the river and (2) the location of the river mouth during the Aptian (especially lower Aptian) robust? Topography, in particular, is generally very difficult to reconstruct in deep-time periods, thus making the reconstruction of watersheds challenging, too. I suggest expanding the arguments supporting the existence of this river, of its watershed such as drawn in Fig. 3 and of the location of the river mouth. If arguments are solid, such discussion will strengthen the manuscript. If they are not, I still believe that the authors' arguments are very interesting but I think it would be good in that case to highlight these uncertainties as a limitation.

**The evolution of the geomorphology and paleo-drainage systems in SW Africa has been studied by numerous authors, who generally acknowledge that two, SW–W-flowing river systems drained most of the interior of southern Africa since the Cretaceous (e.g. De Wit, 1999; Dingle and Hendry, 1984; Partridge and Maud, 1987). Recent apatite fission track data identified a paleo-river valley, dated at 120–110 Ma, which coincides spatially with the present course of the Krom/Sout River that enters**

**the S-Atlantic via the Olifants River near 32° S today (Kounov et al., 2008), roughly 450 km NE of the coring location of Site 361. Based on these results, Kounov et al. (2008) proposed that this deeply incised paleo-valley marks the river course of the paleo-Karoo River, which may have provided "a major outlet to the Atlantic Ocean for its inland drainage system". This interpretation is consistent with onshore geological and morphological evidence (De Wit, 1999). However, we agree that the exact routing of S-African river systems and associated catchment topography during the Early Aptian remains speculative, in particular in the widely eroded upstream regions. To this end, we note that our model results indicate an increase in seasonal precipitation over the entire southern tip of Africa (Supplementary Figure 1), which would cover the assumed catchment area of the paleo-Karoo River.**

**3**. GCM simulation and OAE1a. The authors suggest that a pCO2 doubling from 600 ppm to 1200 ppm is well representative of the climate change that accompanied OAE1a. The choice of the pCO2 values is based on the argument that these values fall "*within the lower range of pCO2 estimates for OAE 1a (Naafs et al., 2016)*" (also lines 160–163). However, different climate models have different climatic sensitivities, i.e., a given change in pCO2 will not have the same effect depending on the climate model in use. Is the climate change simulated in the Kiel model between 600 and 1200 pm really representative of what we know of OAE1a? Interestingly, recent temperature estimates (Naafs and Pancost, 2016; O'Brien et al., 2017) provide an overview of the ocean warming that accompanied OAE1a. I think it would be instructive to provide a rapid comparison between the modeled temperatures and these recent proxy data. Such comparison would satisfactorily complement the interesting comparison of the modeled climatic belts with the lithological indicators of climate already proposed by the authors.

**We agree that a model-data comparison of OAE 1a temperatures would be helpful for the reader to allow an independent validation of the model set-up and results. In fact, we recently published an extensive comparison of our simulated upper ocean temperatures and available proxy records (Steinig et al., 2020), including the data from Naafs and Pancost (2016) and O'Brien et al. (2017). Steinig et al. (2020) used the same model set-up, but with a slightly different South Atlantic and Southern Ocean bathymetry compared to our current study. While the relative temperature increase during OAE 1a is not well constrained due to stratigraphic limitations, we were able to show that absolute OAE 1a proxy temperatures are broadly consistent with our model results at $CO_2$ levels between 1200 to 2400 ppm. This model-data congruence can be achieved by assuming a regional warm bias in the Early Cretaceous $TEX_{86}$ record in the young Atlantic Ocean for which we presented evidence in Steinig et al. (2020). High-resolution atmospheric $CO_2$ reconstructions estimate average levels across OAE 1a of around 1550 ppm (Naafs et al., 2016). We are therefore confident that our modelling approach does reasonably well represent an OAE 1a-like climate state at a $CO_2$ level of 1200 ppm, while the 600 ppm simulation rather reflects the lower Early Cretaceous background $CO_2$ concentrations (Jing and Bainian, 2018). We will add the Steinig et al. (2020) reference and the major results of this study to section 4.4 of the main text to validate our OAE 1a simulation and to provide a broader context on $p$CO$_2$-SST relationships.**

Minor comments:
1. Lines 441–442: the authors state that "*The climate zonation generated by our model indicates that Site 361 and the southern African continent were located in a temperate humid climate belt*". How was this climatic belt identified? Was is through some Köppen classification or similar? Please support this statement.

**We did not apply any quantitative climate zone classification. Instead, based on our modelling results, we located the region of the southern hemisphere westerlies with relatively high precipitation rates compared to the more arid regions in the northern part of the South Atlantic to**

**define the zone of principle change, both in precipitation and zonal atmospheric circulation. We will re-phrase this accordingly in the revised manuscript.**

**2**. Lines 444–446: I suggest including a reference to the modeling study of Chaboureau et al. (2012), which specifically investigated the controlling mechanisms for the deposition of evaporites in the northern and central sections of the nascent South Atlantic Ocean.

**We will reference this closely related study in the revised manuscript. It is important to note that Chaboureau et al. (2012) show a very similar shift towards a positive precipitation minus evaporation balance around the southern tip of Africa.**

**3**. Lines 464–466: I suggest comparing the precipitation simulated at 600 ppm with the precipitation simulated in modern S-Africa in the present-day model control run. This might be more instructive (consistent) than comparing to an independent dataset.

**This is a very helpful suggestion that we will implement in the revised manuscript. The simulated present-day simulation in South Africa amounts to ~2.0 mm/day, which is higher than the observed value of ~1.3 mm/day. The offset is comparable to the CMIP5 multi-model ensemble mean wet bias in this region of ~0.5 - 1 mm/day (Eyring et al., 2016).**

**4**. Lines 496: "*no substantial ocean upwelling occurred along the SW African margin during the Aptian*". Here as well, I suggest approaching the question of the limitations. Indeed, the authors are here reaching the limit of the modeling setup. Coastal upwellings s.s. are 10s of km wide, not 100s. As a consequence, they cannot be captured by the ocean model, the mean horizontal resolution of which is ca. 200 km.

**We fully agree that the model cannot reproduce small-scale local coastal upwelling. However, our results indicate that, unlike today, no large upwelling system (i.e. on the scale of the upwelling system along the west coast of South America; Figure 9a) existed along the SW African coast during the Aptian, as far as our modelling can resolve it. We will acknowledge these limitations and sub-grid-scale upwelling in the revised manuscript.**

**5**. Figs. 9–10: Since the position of Site 361 is critical, please state how its paleocoordinates were calculated and how the point was subsequently located on the mixed Blakey-Sewall map.

**The paleo-coordinates for Site 361 shown in Figure 1 were quantitatively reconstructed using GPlates (Boyden et al., 2011) and the rotational poles published by Matthews et al. (2016). In our model, this paleo-location was qualitatively adapted relative to the model bathymetry. We located the site at the eastern slope of the Cape Basin just off the southern tip of the African continent, consistent with its position on the lower continental rise of South Africa (The Shipboard Scientific Party, 1978). In response to a comment raised by reviewer #2, we added Supplementary Figure 3 showing a detailed map of the implemented model bathymetry. We think this results in a reasonable, qualitative approximation of the 361 paleo-position, especially as we do not discuss any single grid point results for the ocean model.**

Technical comments:
• Line 129: "*The distribution of acyclic hydrocarbons (i.e. n-alkanes, acyclic isoprenoids) were was analyzed*"
• Line 207: "*Distribution of the redox-sensitive trace metals (TMs) Ni, Cu, V, and Zn in black shales are is presented as*"
• Line 417: "*furthermore*" is misleading because the new idea is not used to support the previous paragraph.

• Fig. 2 caption: Missing general title for the Fig.
• Fig. 9 caption: "Positive contours" > positives values (there are no contours).

**We thank reviewer #2 for his technical comments and will include all suggested changes in the revised manuscript.**

[Figure]

**Supplementary Figure 1: (a) Simulated June to August surface wind and total precipitation in the study area at 600 ppm *p*CO₂, (b) relative June to August surface wind and precipitation change due to a doubling of atmospheric *p*CO₂. (c) Monthly mean precipitation and (d) generated surface runoff averaged over the catchment area indicated by the highlighted grid points in (a) and (b). Volume fluxes are integrated over the whole catchment area (~450.000 km2) with annual mean values reported in the respective legends. This figure is similar to Figure 10 of the main text, but the annual mean fields in (a) and (b) are replaced by the respective June to August maps.**

[Figure]

**Supplementary Figure 2:** Surface wind and total precipitation simulated by the KCM at 600 ppm CO₂ (left-hand side panels), 1200 ppm CO₂ (panels in the center) and their respective differences (right-hand side panels). Results are averaged over the whole year (top panels), the months December to February (middle panels) and the months June to August (bottom panels).

[Figure]

**Supplementary Figure 3: Regional ocean model bathymetry used in (a) this study and (b) in the Late Aptian/Early Albian simulations of Dummann et al. (2020). Changed grid points are highlighted in pink. Both studies use the same Kiel Climate Model. To account for the slightly older Aptian time interval investigated in this study, the northward extent of the South Atlantic in (a) was reduced and replaced by the early Aptian boundary conditions from Sewall et al. (2007) northward of ~30°S. Furthermore, the Falkland Plateau (~55°S) is moved to a more eastward position to limit intermediate water exchange between the South Atlantic and Southern Ocean during the early Aptian, consistent with the results of Dummann et al. (2020).**

References cited:

Boyden, J. A., Müller, R. D., Gurnis, M., Torsvik, T. H., Clark, J. A., Turner, M., Ivey-Law, H., Watson, R. J., and Cannon, J. S.: Next-generation plate-tectonic reconstructions using GPlates, in: Geoinformatics: Cyberinfrastructure for the Solid Earth Sciences, edited by: Keller, G. R., and Baru, C., Cambridge University Press, Cambridge, 95-113, 2011.

Chaboureau, A.-C., Donnadieu, Y., Sepulchre, P., Robin, C., Guillocheau, F., and Rohais, S.: The Aptian evaporites of the South Atlantic: a climatic paradox?, Clim Past, 9, 1047-1058, https://doi:10.5194/cp-8-1047-2012, 2012.

De Wit, M.: Post-Gondwana drainage and the development of diamond placers in western South Africa, Economic Geology, 94, 721-740, https://doi.org/10.2113/gsecongeo.94.5.721, 1999.

Dingle, R. V., and Hendry, Q. B.: Late Mesozoic and Tertiary sediment supply to the eastern Cape Basin (SE Atlantic) and palaeo-drainage systems in southwestern Africa, Marine Geology, 56, 13-26, 1984.

Dummann, W., Steinig, S., Hofmann, P., Flögel, S., Osborne, A. H., Frank, M., Herrle, J. O., Bretschneider, L., Sheward, R. M., and Wagner, T.: The impact of Early Cretaceous gateway evolution on ocean circulation and organic carbon burial in the emerging South Atlantic and Southern Ocean basins, Earth Planet Sc Lett, 530, 115890, https://doi.org/10.1016/j.epsl.2019.115890, 2020.

Eyring, V., Righi, M., Lauer, A., Evaldsson, M., Wenzel, S., Jones, C., Anav, A., Andrews, O., Cionni, I., and Davin, E. L.: ESMValTool (v1. 0)–a community diagnostic and performance metrics tool for routine evaluation of Earth system models in CMIP, Geoscientific Model Development, 9, 1747-1802, https://doi:10.5194/gmd-9-1747-2016, 2016.

Kounov, A., Viola, G., De Wit, M., and Andreoli, M.: A Mid Cretaceous paleo-Karoo River valley across the Knersvlakte plain (northwestern coast of South Africa): Evidence from apatite fission-track analysis, South African Journal of Geology, 111, 409-420, https://doi.org/10.2113/gssajg.111.4.409, 2008.

Matthews, K. J., Maloney, K. T., Zahirovic, S., Williams, S. E., Seton, M., and Müller, R. D.: Global plate boundary evolution and kinematics since the late Paleozoic, Global and Planetary Change, 146, 226-250, https://doi.org/10.1016/j.gloplacha.2016.10.002, 2016.

Naafs, B. D. A., Castro, J. M., De Gea, G. A., Quijano, M. L., Schmidt, D. N., and Pancost, R. D.: Gradual and sustained carbon dioxide release during Aptian Oceanic Anoxic Event 1a, Nature Geoscience, 9, 135-139, https://doi.org/10.1038/ngeo2627, 2016.

Naafs, B. D. A., and Pancost, R. D.: Sea-surface temperature evolution across Aptian Oceanic Anoxic Event 1a, Geology, 44, 959-962, https://doi.org/10.1130/g38575.1, 2016.

O'Brien, C. L., Robinson, S. A., Pancost, R. D., Sinninghe Damsté, J. S., Schouten, S., Lunt, D. J., Alsenz, H., Bornemann, A., Bottini, C., and Brassell, S. C.: Cretaceous sea-surface temperature evolution: Constraints from TEX86 and planktonic foraminiferal oxygen isotopes, Earth-Sci Rev, 172, 224-247, https://doi.org/10.1130/abs/2018am-323378, 2017.

Partridge, T. C., and Maud, R. R.: Geomorphic evolution of southern Africa since the Mesozoic, South African Journal of Geology, 90, 179-208, 1987.

Sewall, J. O., van de Wal, R. S. W., van der Zwan, K., van Oosterhout, C., Dijkstra, H. A., and Scotese, C. R.: Climate model boundary conditions for four Cretaceous time slices, Clim Past, 3, 647-657, https://doi.org/10.5194/cpd-3-791-2007, 2007.

Steinig, S., Dummann, W., Park, W., Latif, M., Kusch, S., Hofmann, P., and Flögel, S.: Evidence for a regional warm bias in the Early Cretaceous TEX$_{86}$ record, Earth Planet Sc Lett, 539, 116184, https://doi.org/10.1016/j.epsl.2020.116184, 2020.

The Shipboard Scientific Party: Cape Basin continental rise; Sites 360 and 361, in: Initial Reports of the Deep Sea Drilling Project, Volume 40, edited by: Bolli, H. M., and Ryan, W. B. F., U.S. Government Printing Office, Washington DC (USA), 29-182, https://doi.org/10.2973/dsdp.proc.40.102.1978, 1978.

---

## Author Comment (AC2) · 24 Aug 2020

**Author's response to referee comment (RC) 2**

We would like to thank referee #2 for his/her comments, which we think will significantly improve the quality of our manuscript. Below, we provide a detailed point-to-point response addressing the reviewer's main concerns. New Supplementary Figures 1-6 and a Supplementary Table are referenced in this response and are included at the end of this document.

This manuscript argues that variation between high TOC and modest TOC Aptian black shales at DSDP 361 in the Cape Basin was forced by changes in precipitation patterns over southern Africa. These relatively short-term climatic variations are proposed to be separate from a longer-term tectonic shifts that led organic carbon-rich deposits to be replaced organic carbon-poor deposits from the Aptian through the Albian at this site. Primary evidence cited in support of the climatic interpretation is biomarker analyses, REE distributions, and modelling results. In addition, elemental abundances and other bulk geochemical results are discussed relative to potential diagenetic overprints and redox conditions in the water and sediment column. Data are largely limited to a portion of the lithologies present at one site that seemingly had quite incomplete recovery.

The interpretation is interesting and consistent with the data presented, but to fit the observations, the interpretation does invoke a rather specific set of conditions (e.g., changes in seasonality and geographic focus of precipitation leading to more precipitation and nutrient input from chemical weathering in the interior but more physical weathering along the coast) that are difficult to test. The study also leans heavily on Dummann et al., 2020, EPSL, for data and documentation to the point that I wonder if the last sentence of the first paragraph of the discussion of the EPSL paper refers to the data split off and presented in this manuscript. Regardless of decisions made by authors about whether and how to partition findings among publications, this manuscript would be stronger if the unique contributions of this study were featured and less prominence was given to the broader and longer-term topics addressed in the EPSL paper.

We acknowledge that there is a degree of overlap in content between the present study and our previous study published in EPSL. However, we would like to stress that long-term changes in bottom water redox state related to the opening of South Atlantic-Southern Ocean gateways, targeted in the EPSL publication, played a pivotal role in preconditioning the Early Cretaceous Cape Basin for enhanced organic carbon burial. Different to that, this study focusses on the internal dynamics of organic carbon burial in the Cape Basin, based on diverse and new geochemical data combined with regional climate modeling. Climatic forcing mechanisms acting on shorter time-scales modulated the magnitude of organic carbon burial (i.e. changes between high-TOC and low-TOC black shales), but without an appropriate basin configuration, they were insufficient to generate enhanced organic carbon burial, as stated in e.g. lines 546-547. As such, we consider it important to recapitulate trends in long-term basin evolution to provide a wider context for the fluctuations in organic carbon burial presented and discussed in this study. We are therefore confident that this study presents genuine and new data with a distinct interpretation, which is enriched by reference to other relevant studies.

Emphasizing and expanding presentation of sedimentological data would be one route to this end especially given the importance of deposition patterns and sediment sources in the conclusions. Currently the lithologic data presented are relatively sparse. The paper proposes 3 (or 4) lithostratigraphic units are present that can be differentiated based on color and TOC of the finest grained lithologies. Is there any information on mineralogy, bedding characteristics, or sedimentary structures (physical or trace fossils)?

We fully agree that sedimentology is a central element to understand carbon preservation and we were fortunate to have a large body of sedimentological work already covered in the public literature with a focus on our study site. We consequently decided not to include further

sedimentological analyses into the workflow but rather ensure full coverage of the literature data and interpretation in the revised manuscript. We are confident that the sedimentology presented balances the distinct focus of this study on geochemistry and modeling. Our choice of lithostratigraphic units largely conforms with the units defined by the shipboard scientific party of DSDP Leg 40 (The Shipboard Scientific Party, 1978; Natland, 1978). They placed the boundary between their lithologic units 6 and 7 atop core 28 (The Shipboard Scientific Party, 1978), consistent with the transition from black shale to gray shale deposition proposed in the present study (Figure 2). This transition marks a profound change in the sedimentary environment, as indicated by the appearance of cross-laminated siltstone beds (Natland, 1978; Kagami, 1978) and burrowing structures (The Shipboard Scientific Party, 1978). The shipboard scientists interpreted these lithological changes to reflect a strengthening of deep water circulation and ventilation, respectively, which is consistent with Nd isotope evidence presented in our EPSL paper (Dummann et al., 2020) and a shift towards more oxygenated conditions evident in our new data (cf. lines 312–317, Figure 4, Figure 5). In contrast, black shales in cores 28–48 below (lithostratigraphic unit 7 as defined by the The Shipboard Scientific Party (1978)) are mostly fissile, partly laminated, and lack bioturbation, consistent with a suboxic-anoxic environment (cf. lines 291–311 and Fig. 4). However, it is important to note that gray shales and red beds share similar sedimentary characteristics and thus have been assigned to the same lithological unit 6 by the shipboard scientists. We distinguish them based on their distinct color, differences in TOC content (Figure 2c), S content (Figure 4b), redox-sensitive trace metal enrichment (Figure 4c-f), and Fe-S-TOC relationships (Figure 5), which indicate a shift to fully oxic conditions during the deposition of red beds.

Description of the samples analyzed, their context, and why they were chosen is effectively missing (both for the 'complete set' and the presumably subsets thereof in plots with < than 131 data points).

**The criteria for sample selection are detailed below and will be included in the revised manuscript.**

What about the coarser grained lithologies? It seems these lithologies are an important part of the depositional system, but they are mentioned mostly for having been excluded from consideration. The graphic column, on the other hand, seems to show changes in the proportion of lithologies through time (as well as substantial coring gaps). How does the stratigraphic pattern of different lithologies fit into the provenance/sediment routing interpretation proposed? Petrographic and geochemical examination of the coarser lithologies could provide information about provenance and transport. Do K/Al and Si/Al measures of sedimentologically identified distal turbidites support the assertion that hemipelagic deposition and turbidites can be separated by the metrics used at this site.

Sandstone and sandy mudstones indeed form a substantial part of the sedimentary record, accounting for ~55% of recovered sediment (Figure 2a). These lithologies are interpreted to represent turbidites, dense traction or debris flows, and potentially bed loads, which were deposited in a fan to fan-valley environment (cf. lines 380-381). Detailed information on sandstone petrography, geochemistry, and provenance is provided by Kagami (1978) and Natland (1978). Provenance analyses suggest that sandstones were derived from granitic weathering sources located along the SW African coast (Natland, 1978), indicating a similar weathering source region for sandstones and low-TOC black shales (cf. lines 403 to 410, Figure 3). Long-term changes in relative proportions of the different lithologies have also been discussed by Natland (1978), who proposed that the decreasing abundance of sandstones throughout the Aptian (Figure 2a) was related to the effects of subsidence and erosion of sediment source regions that may have diminished massive coarse-grained turbidites, which characterize the lower part of the black shale interval. Carbon burial in coarse grained sediments is well known to be highly limited and therefore only a secondary focus of our study. At Site 361, TOC contents of sandstones and sandy mudstone beds range from 0.3 – 3% (Erdman and Schorno, 1978) and predominantly comprise terrigenous organic matter, as indicated by Rock-Eval data (Van der Spuy, 2003 and references therein) and the frequent occurrence of wood debris (The Shipboard Scientific Party, 1978). To avoid any bias related to changes in grain-size and to ensure that turbiditic coarse-grained lithologies were omitted, we carefully screened our sample set using geochemical grain-size proxies (i.e. Si/Al and Zr/Al ratios shown in Figure 2e,f; e.g. Croudace and Rothwell (2015) and references therein). These ratios are consistently below or close to average shale values in all chosen samples (Figure 2e,f), and distinct from interbedded sandstones/siltstones, which have been analyzed for reference (Supplementary Figure 1).

However, Si/AI and Zr/AI ratios probably do not allow to readily discriminate between hemipelagic and fine-grained distal turbidites, as they may share similar grain size characteristics. Previous studies support the existence of two genetically distinct (turbiditic and hemipelagic) types of shale (Kagami, 1978; Natland, 1978). Natland (1978), noted that shales frequently comprise finely laminated, varve-like alternations of nannofossil layers and mudstone/shale (Noël and Melguen, 1978). Scanning electron microscopy revealed similar lamination features containing dissolution imprints of coccoliths also in shales in which calcium carbonate has been removed diagenetically (Noël and Melguen, 1978). These shale intervals are up to several meters thick and have been interpreted to be deposited under tranquil hemipelagic conditions (Arthur and Natland, 1979; Natland, 1978). In contrast, grain-size and bedding thickness analyses led Kagami (1978) to conclude that shales associated with sandstone and sandy mudstones beds were probably "deposited by the same sedimentary process" and presumably represent fine-grained turbiditic material, which settled out of suspension. Based on these results, we omitted shale intervals directly on top of sandstone and sandy mudstone during sampling. We further note that  $\delta^{13}C_{org}$  patterns at Site 361 closely track variations in the global ocean-atmosphere reservoir, as discussed in detail in Dummann et al. (2020). This precise reproducibility of  $\delta^{13}$ C trends further supports that only hemipelagic shales were sampled.

How was the 5% organic carbon threshold chosen? Are there really two different lithologies (<5% and >5% Corg) within fine-grained portions of the "black shales" or is there a continuum? What is the stratigraphic character of the most TOC-rich intervals– discrete beds (how thick?) or trends over some thickness (cm? dm? m?). One way to illustrate/test this classification would be a plot of K/Al vs. %organic carbon (cf. Fig. 3b). Statistical tests of apparent geochemical patterns would enhance confidence in the relationships proposed.

Alternations of "high-TOC black shales" (TOC >5%) and "low-TOC black shales" (TOC <5%) appear to be unrelated to changes in lithology and bedding characteristics (i.e. high-TOC intervals are not confined to discrete beds), as illustrated in Supplementary Figure 2a. Instead, increases in TOC content occur over several dm- to m-intervals (considering the relatively low sampling resolution) suggesting transient and gradual shifts in carbon burial, favoring a paleoclimatic/paleoceanographic forcing mechanism. As such, transitions from low-TOC to high-TOC black shale deposition may indeed represent a continuum. This is further supported by steady increases in S/Fe and K/AI ratios with TOC content (Supplementary Figure 2b,c), indicating gradual changes in porewater (and bottom water) oxygenation and sediment (and nutrient) influx from the S-African continent, respectively. However, we argue that the relatively low sampling resolution, which is due to large coring gaps and the scattered occurrence of hemipelagic shales (Figure 2a), does not allow to fully resolve such a continuum. Hence, we make a general distinction between two end member states in terms of redox conditions and burial of organic carbon by using the terms "high TOC black shale" and "low TOC black shale" to convey to the reader the principle differences between these two interrelated units. The exact placement of the TOC threshold along this continuum is debatable. We chose a TOC threshold of 5% based on our paleo-redox analysis, as it appears to mark the transition from suboxic to anoxic-euxinic conditions (Figure 5 and Figure 6). We will discuss these limitations and our choice of TOC threshold in the revised manuscript.

Were there efforts to separate lithogenic from hydrogenic contributions to the REE profiles measured? Could the differences in 3c-d between bulk analyses reflect oceanographic/diagenetic conditions correlated with organic carbon content rather than changes in sediment provenance causing changing in oceanographic/depositional conditions? Some of this information is in the EPSL manuscript and I assume some is in the DSDP volumes cited, but this study would be stronger if more supporting sedimentological and stratigraphic information was summarized and synthesized in the manuscript.

The REE and immobile trace element data presented were obtained by total digestion (cf. lines 102– 112) and thus reflect trends in whole-rock composition. Although scavenging by organic carbon has been shown to contribute substantial amounts of REE to the sediment (up to 20% of the total sedimentary REE content; Abanda and Hannigan (2006)), this process probably did not contribute significantly to the differences in REE distribution observed in this study (Figure 3). To this end, we note that "high-TOC black shales" show overall lower REE concentrations than "low-TOC black shales", which is inconsistent with an enhanced scavenging of REEs by organic carbon. Furthermore, studies from various modern sedimentary environments indicate that REE distributions in sedimentary organic matter are characterized by strong enrichments of MREEs (Freslon et al., 2014). "High-TOC black shales", however, lack a similar MREE enrichment relative to "low-TOC black shales", further supporting that scavenging of REEs by organic carbon had a negligible impact on the bulk sedimentary REE distribution.

Along similar lines, the presentation of the modelling and modelling results is limited although those results arguably provide the only independent assessment of the model proposed. Description of the model (2.4) is brief and the changes in bathymetry between the Dummann et al. 2020 EPSL paper's and the bathymetry used in this paper is not illustrated. The paleogeographic figures in the EPSL paper are superior to the rather simple Fig. 1 in this manuscript. In figures 9 and 10 it would be nice to see a panel that has the 1200 ppm CO2 results shown rather than simply showing 600 ppm results (9b, 10a) and a difference (9c, 10b).

We agree with the reviewer that more space could be given to the presentation of our modeling results. Hence, we will add several supplementary figures to the revised manuscript (Supplementary Figures 3–6), which illustrate the implemented bathymetry, the model spin-up, and the results of individual modelling runs (600 ppm and 1200 ppm CO2). Firstly, a detailed map of the implemented regional model bathymetry, along with a comparison to the Dummann et al. (2020) study, will help to illustrate and justify the small bathymetry differences between both studies (Supplementary Figure 3). Secondly, preempting the next reviewer comment, we will also add a time series of simulated upper and deep ocean temperatures, both globally and for the study area during model spin-up (Supplementary Figure 4). These data clearly show that the model reached a quasi-steady state after about 2000 yr during the 3000 yr long model spin-up. Thirdly, in response to the comments made by Alexandre Pohl (reviewer 1), we will add a short reference to a recently published study demonstrating the capability of the KCM to reproduce available Aptian SST proxy data (Steinig et al., 2020). References to these additional figures and studies will be included in section 2.4 of the revised manuscript, which will allow a more in-depth assessment of the model results by the reader. Finally, we agree that some readers might be interested in a more detailed presentation of the modeling results. However, we argue that the current versions of Figure 9 and 10 provide a balance between simplicity and the main results of the modeling. Hence, we prefer to include the current version of Figure 9 and, in response to the comments made by reviewer 1, we will include a modified version of Figure 10 in the main text, which shows precipitation pattern during austral summer (cf. Supplementary Figure 6g,i). Detailed results of the individual simulations (Supplementary Figures 5 and 6) will, however, be provided in the supplement, using the same quantities and contour intervals as in Figures 9 and 10. All supplemental figures will be referenced in section 4.4 of the main text.

The idea that an intermediate water mass isolates the deep Cape Basin promoting organic carbon accumulation is an interesting model-based suggestion. Could cross sections across the basin showing

model predicted water properties that indication a saline layer divides the water column be provided as well as sensitivity tests of the persistence and extent of such a layer under the two climatic modes proposed to explain the distribution of TOC in the black shales? Could discussion and supporting graphics demonstrating the models have stabilized be provided especially relative to the deep basin?

We agree that the impact of deep and intermediate water mass circulation on organic carbon burial is an interesting and relevant question. Changes in ocean circulation are, however, difficult to assess based on the available data, which renders it difficult to validate circulation model results. We therefore think that such a discussion is beyond the scope of the present proxy-based study. However, a follow-up modeling study with a distinct focus on the sensitivity of the South Atlantic circulation system towards climatic (i.e. changes  $pCO_2$ ) and tectonic forcing (i.e. opening of gateways) processes is currently in preparation. The model equilibrium is illustrated in Supplementary Figure 4 as described in our response to the previous comment.

Finally, the manuscript suggests that increased CO2 might be the cause of the proposed changes in precipitation leading to nutrient input and enhance physical weathering proposed to explain organic carbon enrichment associated with OAE1a. The manuscript also suggests orbital variations might be the cause of the other organic carbon rich black shale intervals. Are the authors proposing that orbital variations cause a change in CO2? If so, how? If they instead are suggesting both orbital variations and CO2 increase cause the same climatic changes, this would be an idea that seems worth including in section 4.6. Would the authors hypothesize what orbital configuration would be expected to mimic high CO2 precipitation patterns and why? Such a prediction could be tested when/if an orbital timescale is generated for the Aptian Cape Basin.

In line with previous studies (e.g. Naafs et al., 2016), we invoke an increase in  $pCO_2$  as a primary forcing mechanism to explain changes in continental hydrology and marine organic carbon burial during OAE 1a, while orbital forcing is considered as an additional modulating mechanism, with different mechanisms and impacts on OC production and burial depending on the study region. We do not claim that orbital scale fluctuations directly impact atmospheric pCO2, however, over longer time scales orbital forcing responds to fluctuations in pCO2. Wagner et al. (2013) addressed these relationships by presenting a conceptual model that links marine organic carbon burial dynamics to changes in atmospheric circulation and continental hydrology under the descending/ascending limbs of the atmospheric Hadley Cell. They proposed that orbitally driven contraction/expansion of the Hadley Cell caused latitudinal shifts of climate belts, inducing cyclic fluctuations in continental river run-off and oceanic upwelling and thereby variations in marine organic carbon burial. Given the position of Site 361 close to the descending limb of the southern hemisphere Hadley Cell (Figure 9 and 10), we speculate that the episodic occurrence of high-TOC black shales in the Cape Basin may have been related to a recurrent northward migration of the austral westerlies due to contraction of the Hadley Cell. This may have increased humidity and precipitation in the S-African hinterland, augmenting continent-ocean nutrient transfer and ultimately organic carbon burial in the Cape Basin. However, we are currently limited in providing reliable estimates of orbital cyclicity forcing high-TOC black shale formation due to the overall low stratigraphic coverage at Site 361 (Fig. 2a).

55- does the mid-ocean ridge represent a barrier to circulation?

The role of the mid-ocean ridge for bottom water circulation (i.e. >1000 m water depth) is currently unclear, as no Early Cretaceous sediment records are available from the deepest part of the Argentine Basin west of the ridge. However, it probably did not represent a barrier to intermediate water mass circulation (i.e. <1000 m water depth), as detailed in Dummann et al. (2020).

57- how is global excess OC burial defined/calculated

McAnena et al. (2013) used biogeochemical modeling to calculate the amount of excess organic carbon buried that is required to explain a positive  $\delta^{13}$ C excursion during the Late Aptian, which was associated with a cooling event (the Late Aptian Cold Snap; LACS). The modeling results suggest that an excess of ~812,000 gigatons of carbon may have been buried over a time span of 2.5 million years, about 16% of which may have buried in the emerging South Atlantic. Technical details on the modeling approach can be found in McAnena et al. (2013).

59- Angola Basin not shown in Figure

**We will add the names of the different basins to Figure 1.**

63- which data- TOC percentages?

**The study of Behrooz et al. (2018) is primarily based on biomarker data.**

87- beyond TOC (and color?) are there sedimentological differences among lithostratigraphic divisions, gray shales and low-TOC black shales do not seem different based on the criteria used 95- describe sample set

**Our sampling strategy and choice of lithostratigraphic units is outlined above and will be detailed in section 2.1 (cf. line 80) of our revised manuscript.**

126, 235- don't start sentence with lowercase letter, write around

**We will re-phrase these sentences accordingly.**

144- not sure this is a method, odd to exclude sandstones for provenance work

**As mentioned above, detailed information on the provenance of sandstones can be found in the literature (e.g. Natland, 1978).**

149- These data

**Will be amended as suggested.**

177- very different precision suggested for characteristic AS Si/Al and Zr/Al ratios

Average shale values are taken from the literature (Wedepohl, 2004, 1971). The different precisions reported for major (Si) and minor/trace elements (Zr) most likely depend on the analytical method chosen by these authors.

180- difference seems small given variance- is it statistically supported. What does a plot of TOC vs. K/Al look like?

**See Supplementary Figure 1b.**

254- is proposed difference statistically supported?

To evaluate if the differences in the fractional abundances of desmethylsteranes are statistically significant, we performed a t-test (p=0.05). The results, summarized in Supplementary Table 1, indicate that high-TOC black shales contain significantly higher mean abundances of C27- and C28- desmethylsteranes, reflecting inputs from marine sources (Huang and Meinschein, 1979), and lower

**abundances of $C_{29}$ -desmethylsteranes, commonly attributed to higher land plant inputs (Huang and Meinschein, 1979), compared to low-TOC black shales.**

Figure 2- using red shading for organic carbon rich black shales and orange shading for red beds seems like an odd choice; panels d-f are labeled grain size/mineralogy, but the plots show elemental ratios.

We will change the color shading for "high-TOC black shales". The labels in the different panels of Figure 2d–f intend to clarify our interpretation of the individual geochemical proxies (i.e. Zr/Al and Si/Al are used as proxies for grain-size, K/Al is used as proxy for changes in clay mineralogy).

Supplementary information could include some description of the plots.

Descriptions will be added alongside supplementary information on our modeling approach.

|                                      | C27-              | C28-              | C20-              |
|--------------------------------------|-------------------|-------------------|-------------------|
|                                      | desmethylsteranes | desmethylsteranes | desmethylsteranes |
|                                      | [%]               | [%]               | [%]               |
| x LTBS                               | 25.4              | 33.5              | 41.1              |
| x HTBS                        | 28.6              | 39.1              | 32.3              |
| σ 2 LTBS                  | 11.9              | 26.2              | 44.5              |
| σ² HTBS                              | 22.2              | 30.2              | 32.1              |
| Hypothetical difference of means     | 0                 | 0                 | 0                 |
| T-statistics                         | -3.1              | -4.1              | 5.8               |
| Critical t-value (one-tailed t-test) | 1.7               | 1.7               | 1.7               |
| Critical t-value (two-tailed t-test) | 2.0               | 2.0               | 2.0               |

**Supplementary Table 1: T-test results (p=0.05) for the fractional abundances of C27–C29-desmethylsteranes (LTBS=low-TOC black shales, HTBS=high-TOC black shales)**

Supplementary Figure 1: Cross-plot showing Si/Al and Zr/Al ratios for high-TOC black shales (TOC>5%), low-TOC black shales (TOC

---

## Author Response (AR1)

**University of Cologne**

[Figure]

University of Cologne • Albertus-Magnus-Platz • 50923 Köln

**Faculty of Mathematics and Natural Sciences**

**Institute of Geology and Mineralogy**

Wolf Dummann
(wdummann@uni-koeln.de)

Cologne, 20 November 2020

Dear Prof. Donnadieu, thank you for considering our research article entitled " Driving mechanisms of organic carbon burial in the Early Cretaceous South Atlantic Cape Basin (DSDP Site 361)" for publication in Climate of the Past. We are happy to share our revised version of manuscript cp-2020-15, which has been modified according to your suggestions and the suggestions made by two reviewers. We are confident that we have addressed all of the reviewers' concerns and outline all major changes below.

The revised version of the manuscript puts more emphasis on short-term organic carbon (OC) burial variability, while we reduced our discussion on long-term (tectonic) trends to avoid overlap with our previous study published in EPSL. However, we decided to include a brief discussion on long-term redox changes (chapter 4.2.2) to provide the reader with a wider paleoceanographic and tectonic context for the data presented and discussed in this study. To test the impact of orbital forcing on OC burial, we conducted a series of new model simulations implementing different orbital configuration (i.e. variations in precession and obliquity). Our GCM results indicate that orbital forcing exerts a profound control over continental hydrology in southern Africa, as reflected in marked changes in precipitation and total run-off. Importantly, the magnitude of run-off changes due to orbital forcing is greater than the run-off change induced by a doubling of $p$CO$_2$. Based on these results, we hypothesize that orbital variations, specifically minima in obliquity and a perihelion during southern winter solstice, were the main driver of short-term fluctuations in OC burial in the Cape Basin. This interpretation is consistent with our proxy data, which indicate that conditions conducive to enhanced burial of OC recurred episodically before, during, and after OAE 1a. To enable the reader to assess our new and previous model results in-depth, we included a total of 7 supplementary figures (Supplement S4), as suggested by reviewer 2.

To address the other major points of criticism of reviewer 2, we included background information on the sedimentology of shales and coarse-grained turbiditic sediments (chapters 1.1 and 2.1), detailed our choice of lithostratigraphic (sub-)units (chapters 1.1 and 3.1), described our sampling approach (chapter 2.1), and discuss the impact of deep water circulation on OC burial (lines 350-357). For this, we provide T and S latitudinal-depth transects across the southern sector of the S-Atlantic (Supplementary Figure 3 in Supplement S4).

Zülpicher Straße 49a
50764 Cologne
Telefon +49 221 470-7316
Telefax +49 221 470-1663

In response to the comments made by reviewer 1, we note that our new modeling data suggest that humidity/aridity changes in response to orbital forcing occurred over the entire continental area of southern Africa (i.e. coastal regions and the continent's interior), which resolves the apparent discrepancy between model and proxy data mentioned by reviewer 1 and reduces the influence of uncertainties in the exact placement of the proposed catchment area for the significance of our results. We further included a reference to a previous study (Steinig et al., 2020), which showed that the KCM is capable of simulating upper ocean temperatures at a $p$CO$_2$ of 1200 ppm that are consistent with proxy reconstructions for OAE 1a.

Please find a version of the revised manuscript with marked changes below. We would like to thank all reviewers again for their constructive feedback on our original submission, which was very helpful in improving the quality of our paper. We would be glad to respond to any further questions and comments that you may have.

Sincerely,

Wolf Dummann

[revised manuscript text omitted]